# TRANSFORMERS LEARN BAYESIAN NETWORKS AUTOREGRESSIVELY IN-CONTEXT

## ABSTRACT

Transformers have achieved tremendous successes in various fields, notably excelling in tasks involving sequential data like natural language processing. Despite their achievements, there is limited understanding of the theoretical capabilities of transformers. In this paper, we theoretically investigate the capability of transformers to autoregressively learn Bayesian networks in-context. Specifically, we consider a setting where a set of independent samples generated from a Bayesian network are observed and form a context. We show that, there exists a simple transformer model that can (i) estimate the conditional probabilities of the Bayesian network according to the context, and (ii) autoregressively generate a new sample according to the Bayesian network with estimated conditional probabilities. We further demonstrate in extensive experiments that such a transformer does not only exist in theory, but can also be effectively obtained through training. Our analysis showcases the potential of transformers to effectively learn complicated probabilistic models, and contributes to a better understanding of the success of large language models.

## 1 INTRODUCTION

Transformers (Vaswani et al., 2017) have achieved tremendous success across various fields. These models have particularly revolutionized the way we approach problems related to text generation, translation, and understanding by leveraging their ability to capture long-range dependencies and contextual information. Despite these achievements, there remains limited understanding of the theoretical capabilities of transformers.

To theoretically understand the success of transformers, a notable line of recent works (Akyürek et al., 2022; Zhang et al., 2023; Bai et al., 2023; Huang et al., 2023; Nichani et al., 2024) have studied the power of transformers in solving in-context learning tasks. Specifically, Akyürek et al. (2022); Zhang et al. (2023); Bai et al. (2023); Huang et al. (2023) theoretically studied how transformers can perform in-context linear regression under the setting that the context consists of a training data set and the query token contains a test data for prediction. More recently, Huang et al. (2023) studied how transformers can learn causal structures. Specifically, under the assumption that the tokens consist of multiple sequences of samples generated from a causal network, Huang et al. (2023) demonstrated that gradient descent can pre-train a transformer to learn the causal network structure. By doing so, when transformer sees a new context-query pair, it can generate prediction according to the learned network structure and the context. However, the analysis in Huang et al. (2023) was mostly limited to the setting where each variable has at most one parent.

In this work, we focus on a specific setting where a set of independent samples generated from a Bayesian network are observed and form a context, and our goal is to investigate the capability of transformers to autoregressively learn Bayesian networks in-context. The main contributions of this paper are two-fold: providing clean and intuitive theoretical proofs, and presenting robust empirical studies. Specifically, our contributions can be summarized as follows.

- Theoretically, we demonstrate the existence of a transformer model that is capable of: (i) estimating the conditional probabilities of the Bayesian network given the context, and (ii) autoregressively generating a new sample based on these estimated conditional probabilities. This gives an intuitive demonstration on the capability of transformers to perform complicated sampling tasks.

- Empirically, we perform extensive experiments to validate our theoretical claims. Specifically, under various settings where the Bayesian network is a (Markov) chain, a tree, or a general graph, we demonstrate that a transformer can indeed be pre-trained from scratch, so that it can perform in-context estimations of conditional probabilities, and help sample a new sequence of variables accordingly.

**Notations.** We use lowercase letters to denote scalars and boldface lowercase/uppercase letters to denote vectors/matrices, respectively. For a matrix $\mathbf{A}$, we use $\|\mathbf{A}\|_2$ to denote its spectral norm. For an integer $n$, we denote $[n] = 1, 2, \ldots, n$. For a set $S$, we use $|S|$ to denote its cardinality. We also use $\mathbb{1}[\cdot]$ to denote an indicator function that equals 1 when the corresponding statement is true and equals 0 otherwise.

## 2 RELATED WORK

**Transformers.** Transformers Vaswani et al. (2017) and its variants have demonstrated its success in various of domains such as language Devlin (2018); Liu (2019); Raffel et al. (2020); Touvron et al. (2023); Achiam et al. (2023), vision Dosovitskiy (2020); Jia et al. (2022); Liu et al. (2021); Peebles & Xie (2023), multi-modality Gal et al. (2022); Radford et al. (2021); Li et al. (2023a) etc. Large language models (LLMs) demonstrate remarkable ability to learn tasks in-context during inference, bypassing the need to update parameters Brown (2020); Lampinen et al. (2022); Khandelwal et al. (2018). However, the understanding of the inner mechanisms of these models, and how they perform such complex reasoning tasks is largely remain undiscovered (Dong et al., 2022). Such disadvantage prevents us to interpret why transformers often struggles to generalize well under out-of-distribution scenarios, especially on simple reasoning and logical tasks such as arithmetic (Magister et al., 2022; Touvron et al., 2023; Ebrahimi et al., 2020; Suzgun et al., 2022). This raise a doubt on how and when can transformers learn the appropriate algorithms to solve tasks or not.

**In-Context Learning.** Recently, a line of work studies transformers through the lens of in-context learning (ICL) (Dong et al., 2022; Zhang et al., 2023), an ability of models to generate predictions based on a series of examples. Empirically, recent studies find out transformers are capable of learning a series of functions in-context (Garg et al., 2022; Wei et al., 2023; Zhang et al., 2023; Zhou et al., 2023; Grazzi et al., 2024; Park et al., 2024; Akyürek et al., 2022), showing transformers can learn to approximate a wide range of algorithms. Theoretically, several works also analyze the algorithmic approximation perspective of transformers under various of conditions (Von Oswald et al., 2023; Nichani et al., 2024; Shen et al., 2023; Ahn et al., 2024; Li et al., 2023b; Wies et al., 2024). A recent work Von Oswald et al. (2023) shows that linear transformers (Katharopoulos et al., 2020) are capable of performing gradient descent based on in-context examples. In (Bai et al., 2023), they not only show ReLU transformers are capable of approximating gradient descent with small error, but can also capable of implementing more complex ICL processes involving *in-context algorithm selection*. To the best of authors' knowledge there is no existing literature that theoretically and empirically shows transformers learn Bayesian network in-context.

## 3 IN-CONTEXT LEARNING OF BAYESIAN NETWORKS

In this section, we briefly review the definitions of Bayesian network models, and formally define the problem of learning Bayesian networks in-context.

**Bayesian networks.** A Bayesian network is a probabilistic graphical model which specifies the conditional dependencies among the variables by a directed acyclic graph. Each node of the Bayesian network represents for a random variable, and the edges connected to a node indicates the "parent(s)" and "child(ren)" of the node. Furthermore, Bayesian networks modeling discrete random variables can be parameterized by parameters that form *conditional probability tables*, which define the conditional distribution of each random variable given its parent(s).

**Learning of Bayesian networks autoregressively in-context.** Consider $M$ discrete random variables $X_1, \ldots, X_M$. Then there exists a Bayesian network $\mathcal{B}$ modeling the joint distribution of $X_1, \ldots, X_M$, which satisfies that $X_1$ is a root variable with no parents, and that for any $i \in [M]$, the parents of $X_i$ are with indices smaller than $i$. For a given Bayesian network $\mathcal{B}$, and for any $m \in [M]$,

we denote by $\mathcal{P}(m)$ the index set of all parents of $X_m$. Then the *maximum in-degree* of $\mathcal{B}$ can be defined as $\max_{m \in [M]} |\mathcal{P}(m)|$.

In this work, we consider the problem of learning the conditional probability tables of such a Bayesian network $\mathcal{B}$ in-context. Specifically, suppose that we are given $N$ groups of context observations $X_{1i}, \ldots, X_{Mi}, i = 1, \ldots, N$ that are independently generated according to $\mathcal{B}$. Then, we study the capability of transformers to autoregressively sample a new sequence $X_{1q}, \ldots, X_{Mq}$ based on conditional probability tables estimated from the context.

Suppose that the discrete random variables $X_1, \ldots, X_M$ takes $d$ possible values, and for $i \in [N]$ and $m \in [M]$, denote by $\mathbf{x}_{mi}$ the one-hot vector of the observation $X_{mi}$. Moreover, suppose that at a certain step during the autoregressive generation process, some variables among $X_1, \ldots, X_M$ have been generated, and the goal is to generate the next variable. We define the query sequence $\mathbf{x}_{1q}, \ldots, \mathbf{x}_{Mq}$ as follows:

- If $X_{mq}$ is already sampled, then $\mathbf{x}_{mq}$ is the one-hot vector representing the obtained value.

- If $X_{mq}$ is not sampled, then $\mathbf{x}_{mq}$ is a zero vector.

Suppose that at the current step, the target is to sample $X_{m_0 q}$. We define additional vectors

$$\mathbf{p} = [\mathbf{0}_{d(m_0-1)}^\top, \mathbf{1}_d^\top, \mathbf{0}_{d(M-m_0+1)}^\top]^\top, \ \mathbf{p}_q = [\mathbf{0}_{d(m_0-1)}^\top, \mathbf{1}_d^\top, \mathbf{0}_{d(M-m_0)}^\top, \mathbf{1}_d]^\top \in \mathbb{R}^{(M+1)d}. \quad (3.1)$$

The definition of $\mathbf{p}$ and $\mathbf{p}_q$ serves two purposes. First of all, they can teach an autoregressive model the current variable-of-interest. Moreover, the difference bewteen $\mathbf{p}$ and $\mathbf{p}_q$ also serves as an indicator of the "query" variable in the input. Based on these definitions, we define

$$\mathbf{X} = \begin{bmatrix} \mathbf{x}_{11} & \mathbf{x}_{12} & \cdots & \mathbf{x}_{1N} & \mathbf{x}_{1q} \\ \mathbf{x}_{21} & \mathbf{x}_{22} & \cdots & \mathbf{x}_{2N} & \mathbf{x}_{2q} \\ \vdots & \vdots & & \vdots & \vdots \\ \mathbf{x}_{M1} & \mathbf{x}_{M2} & \cdots & \mathbf{x}_{MN} & \mathbf{x}_{Mq} \\ \mathbf{p} & \mathbf{p} & \cdots & \mathbf{p} & \mathbf{p}_q \end{bmatrix}, \quad (3.2)$$

The matrix $\mathbf{X}$ can then be directly fed into a transformer model whose output aims to give the estimated distribution of $X_{m_0 q}$ as a $d$-dimensional vector that sums to one. If such a transformer model exists, then the autoregressive sampling process can be achieved according to Algorithm 1. The major goal of this paper is to investigate whether transformers can handle such tasks well.

---

**Algorithm 1** Autoregressive Sampling

1: **input:** Observations $\{\mathbf{x}_{mi} : m \in [M], i \in [N]\}$, model $\mathbf{f} : \mathbb{R}^{(2M+1)d \times (N+1)} \to \mathbb{R}^d$.
2: Initialize $\mathbf{x}_{mq} = \mathbf{0}_d$ for $m \in [M]$.
3: **for** $m_0 = 1$ **to** $M$ **do**
4:     Set $\mathbf{p}$ and $\mathbf{p}_q$ according to equation 3.1, and define $\mathbf{X}$ according to equation 3.2.
5:     Sample $X_{m_0 q}$ according to $\mathbf{f}(\mathbf{X})$, and update $\mathbf{x}_{m_0 q}$ as the corresponding one-hot vector.
6: **end for**

---

**Optimal maximum likelihood estimation of conditional probabilities.** To measure the performance of transformers, we consider comparing the output of the transformer with the optimal conditional distribution estimation given by maximizing the likelihood. For discrete random variables, it is will-known that the maximum likelihood estimation is obtained by *frequency counting*. Specifically, suppose that at a certain step in the autoregressive sampling procedure, the model is aiming to sample the $m_0$-th variable. Then, the optimal sampling probability vector $\mathbf{p}_{m_0}^{\mathrm{opt}} \in \mathbb{R}^d$ is given by

$$[\mathbf{p}_{m_0}^{\mathrm{opt}}]_j = \frac{|\{i \in [N] : X_{m_0 i} = j, \text{ and } X_{mi} = X_{mq} \text{ for all } m \in \mathcal{P}(m_0)\}|}{|\{i \in [N] : X_{mi} = X_{mq} \text{ for all } m \in \mathcal{P}(m_0)\}|}.$$

Further by the fact that $\mathbf{x}_{mi}$'s and $\mathbf{x}_{mq}$'s are one-hot vectors, we can also write

$$\mathbf{p}_{m_0}^{\mathrm{opt}} = \sum_{i \in [N]} \frac{\mathbb{1}[\mathbf{x}_{mi} = \mathbf{x}_{mq} \text{ for all } m \in \mathcal{P}(m_0)]}{|\{i \in [N] : \mathbf{x}_{mi} = \mathbf{x}_{mq} \text{ for all } m \in \mathcal{P}(m_0)\}|} \cdot \mathbf{x}_{m_0 i}.$$

To compare a function output $\mathbf{f} \in \mathbb{R}^d$ with the optimal solution above, we consider the total variation distance between the two corresponding distributions, which is defined as

$$\mathrm{TV}(\mathbf{f}, \mathbf{p}_{m_0}^{\mathrm{opt}}) := \frac{1}{2} \sum_{j=1}^{d} |[\mathbf{f}]_j - [\mathbf{p}_{m_0}^{\mathrm{opt}}]_j|.$$

## 4 MAIN THEORY

We consider standard transformer architectures introduced in Vaswani et al. (2017) that consists of self-attention layers and feed-forward layers with skip connections. Specifically, in our setup, an attention layer with parameter matrices $\mathbf{V} \in \mathbb{R}^{(2M+1)d \times (2M+1)d}, \mathbf{K} \in \mathbb{R}^{Md \times (2M+1)d}, \mathbf{Q} \in \mathbb{R}^{Md \times (2M+1)d}$ is defined as follows:

$$\mathrm{Attn}_{\mathbf{V},\mathbf{K},\mathbf{Q}}(\mathbf{X}) = \mathbf{X} + \mathbf{V}\mathbf{X}\mathrm{softmax}[(\mathbf{K}\mathbf{X})^\top (\mathbf{Q}\mathbf{X})],$$

where $\mathrm{softmax}$ denotes the column-wise softmax function. In addition, an feed-forward layer with parameter matrices $\mathbf{W}_1, \mathbf{W}_2 \in \mathbb{R}^{(2M+1)d \times (2M+1)d}$ is defined as follows:

$$\mathrm{FF}_{\mathbf{W}_1,\mathbf{W}_2}(\mathbf{X}) = \mathbf{X} + \mathbf{W}_2\sigma(\mathbf{W}_1\mathbf{X}),$$

where $\sigma(\cdot)$ denotes the entry-wise activation function. We consider the ReLU activation function $\sigma(z) = \max\{0, z\}$. Given the above definitions, we follow the convention in Bai et al. (2023) and call the following mapping a "transformer layer":

$$\mathrm{TF}_{\boldsymbol{\theta}}(\mathbf{X}) = \mathrm{FF}_{\mathbf{W}_1,\mathbf{W}_2}[\mathrm{Attn}_{\mathbf{V},\mathbf{K},\mathbf{Q}}(\mathbf{X})],$$

where $\boldsymbol{\theta} = (\mathbf{V}, \mathbf{K}, \mathbf{Q}, \mathbf{W}_1, \mathbf{W}_2)$ denotes the collection of all parameters in the self-attention and feed-forward layer.

The above specifies the definition of a transformer layer, which is a mapping from $\mathbb{R}^{(2M+1)d \times (N+1)}$ to $\mathbb{R}^{(2M+1)d \times (N+1)}$ (for any $N \in \mathbb{N}_+$). To handle the task of generating $d$-class categorical variables, we also need to specific the output of the model, which maps matrices in $\mathbb{R}^{(2M+1)d \times (N+1)}$ to vectors in $\mathbb{R}^d$. Here we follow the common practice, and define the following $\mathrm{Read}(\cdot)$ function

$$\mathrm{Read}(\mathbf{Z}) := \mathbf{Z}\mathbf{e}_{N+1} \text{ for all } \mathbf{Z} \in \mathbb{R}^{(2M+1)d \times (N+1)}$$

to output the last column of the input matrix, and consider a linear mapping $\mathrm{Linear}_{\mathbf{A}}(\cdot)$ to convert the output of the $\mathrm{Read}(\cdot)$ function to the final distribution vector:

$$\mathrm{Linear}_{\mathbf{A}}(\mathbf{z}) = \mathbf{A}\mathbf{z} \text{ for all } \mathbf{z} \in \mathbb{R}^{(2M+1)d},$$

where $\mathbf{A} \in \mathbb{R}^{d \times (2M+1)d}$ is the paramter matrix of the linear mapping.

Given the above definitions, we are ready to introduce our main theoretical results, which are summarized in the following theorem.

**Theorem 4.1.** For any $\epsilon > 0$, and any Bayesian network $\mathcal{B}$ with maximum in-degree $D$, there exists a two-layer transformer model

$$\mathbf{f}(\mathbf{X}) = \mathrm{Linear}_{\mathbf{A}}[\mathrm{Read}(\mathrm{TF}_{\boldsymbol{\theta}^{(2)}}(\mathrm{TF}_{\boldsymbol{\theta}^{(1)}}(\mathbf{X})))]$$

with parameters satisfying

$$\|\mathbf{V}^{(1)}\|_2, \|\mathbf{K}^{(1)}\|_2, \|\mathbf{Q}^{(1)}\|_2, \|\mathbf{W}_2^{(1)}\|_2, \|\mathbf{V}^{(2)}\|_2, \|\mathbf{W}_1^{(2)}\|_2, \|\mathbf{W}_2^{(2)}\|_2, \|\mathbf{A}\|_2 \le 1,$$
$$\|\mathbf{W}_1^{(1)}\|_2 \le 2\sqrt{D+1}, \|\mathbf{K}^{(2)}\|_2, \|\mathbf{Q}^{(2)}\|_2 \le 3\log(MdN/\epsilon),$$

such that for any $m_0 \in [M]$ and $\mathbf{p}, \mathbf{p}_q$ defined according to $m_0$, it holds that

$$\mathrm{TV}\{\mathbf{f}(\mathbf{X}), \mathbf{p}_{m_0}^{\mathrm{opt}}\} \le \epsilon.$$

Theorem 4.1 shows that there exists a two-layer transformer with an appropriate linear prediction layer such that, for any variable of interest $X_{m_0}$, the transformer can always output a distribution vector that is close to the optimal maximum likelihood estimation $\mathbf{p}_{m_0}^{\mathrm{opt}}$ in total variation distance.

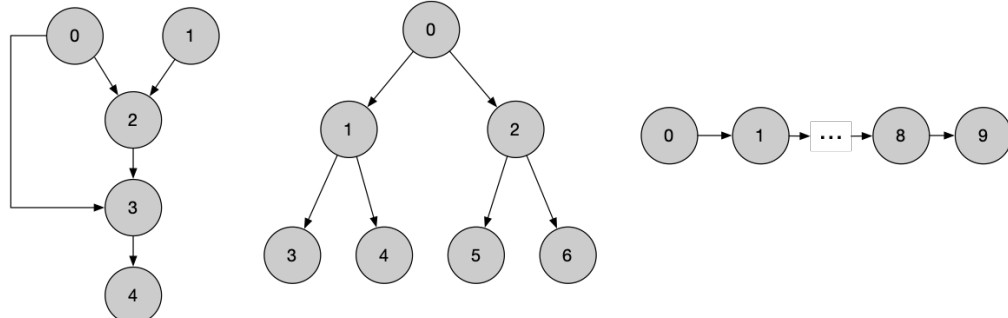

Figure 1: **Illustrations of graph structures in the experiments.** Left to right: general graph, tree and chain. The curriculum follows the number order of variables. The arrow indicates the causal relationships between variables. Note that for general graph, variable 2, 3 both have 2 parents. However, modeling variable 0, 1, 2 is identical for naive Bayes and Bayesian inference, and is **NOT** for variable 3. For tree, modeling root and level-1 variables is identical for naive Bayes and Bayesian inference. For chain, modeling variable 0, 1 is identical for naive Bayes and Bayesian inference.

Importantly, for Bayesian networks with bounded maximum in-degrees, the transformer we demonstrate has weight matrices with bounded (up to logarithmic factors) spectral norms. This provides strong evidence of the efficiency (Bai et al., 2023) of transformers in learning Bayesian networks in-context.

A notable pattern of the result in Theorem 4.1 is that it demonstrates the capability of transformers to generate a sequence of variables in an autoregressive manner – the parameters of the transformer do not depend on the index of the variable of interest $m_0$, and the same transformer model works for all $m_0 \in [M]$ as long as the vectors $\mathbf{p}$, $\mathbf{p}_q$ appropriately defined according to $m_0$. This means that, the transformer model $\mathbf{f}(\mathbf{X})$ can be utilized in the autoregressive sampling procedure in Algorithm 1, such that at each step, the transformer always sample the corresponding variable with close-to-optimal distributions.

## 5 EXPERIMENTS

The main paper contains three parts of the experiments. First, we verify our theoretical results by studying the capabilities of transformers learning Bayesian networks. Second, we analyze whether trained transformers are capable of generalizing to different value of $N$. Last, we perform a case study on whether our theoretical construction is optimal. In the appendix, we show the impact of different parameters on model performance.

### 5.1 TRANSFORMERS LEARN BAYESIAN INFERENCE

Here we conduct the experiment of training transformers to perform Bayesian inference. We also visualize their convergence result with loss and accuracy curves.

**Datasets.** We consider training transformers to learn Bayesian networks of three structures: chain, tree and general graph, see Figure 1 for illustration. All variables in our dataset are with binary values (2 possible outcomes). For each structure, we generate 50k graphs with randomly initialized probability distributions, and sample all training data from them. To reduce noise during training, the probability distributions of those graphs are sampled from one of the following uniform distributions $\{U(0.15, 0.3),\ U(0.7, 0.85)\}$. Specifically, we try to avoid two scenarios: (i) variables are independent to each other (probability close to 0.5) and (2) deterministic relationship between variables (probability close to 0 or 1), as these scenarios either lose the graph structure or the probabilistic nature of networks. Empirically, we also find out models converge better under this approach.

**Metrics.** We denote the number of examples during training as $N_{\text{train}}$, and $N_{\text{test}}$ as the number of examples during evaluation. For evaluations, we randomly generate 1 graph for each graph structure as testset. We report the accuracy of transformers, Naive Bayes, Bayesian inference and the optimal accuracy on testset and vary the number of examples in each prediction. Note that the optimal accuracy is not 1 due to the probabilistic nature of networks. For each number of examples $N_{\text{test}} \in$

[5, 100], we randomly sample a set of 1500 observations, with each observation contains $N_{\text{test}}$ ICL examples and 1 test token. We separate the evaluation of each variables in the graph as they have different optimal accuracy. The reported accuracy are the average over 10 runs with different random seeds. Due to space limit, we select 3 variables for each graph structure to present. Experimental details are in Appendix C.2.

**Model.** We follow most settings in Bai et al. (2023) by using a transformer with 8 heads and 256 hidden dimension for all experiments, the only difference is we use 6 instead of 12 layer transformers. A small difference is we set $\boldsymbol{p} \in \mathbb{R}^M$ as zero vectors, and $\boldsymbol{p}_q \in \mathbb{R}^M$ a one-hot vector, indicating the variable to predict. Note that we remove the positional embedding/encoding from the transformer to highlight the utility of $\boldsymbol{p}$. The input dimension is always 3 times of the number of variables in the graph as our construction in Equation 3.2.

**Setup.** We set $N_{\text{train}} = 100$ during training, and evaluate models on different size of $N_{\text{test}}$. Similar to Bai et al. (2023), we train the transformer with carefully designed curriculum discussed in the following paragraph. After the training loss reach to a threshold, we then advance the curriculum by revealing one extra variable. We find out this approach helps models to learn the graph structure better. We explain details the in following paragraph.

**Curriculum Design.** We take the data-level curriculum approach to train the transformers on Bayesian Inference. The goal of the curriculum is to lead transformers to learn the whole graph structure well. We determine the difficulty of the curriculum by the number of variables in the graph. Therefore, we design the curriculum from easy to hard by revealing more and more variables throughout training. By doing so, the graph structure "grows" during training. We start by revealing only the first two variables in the graph, meaning the transformer will only learn to predict the first 2 variables. After training loss reaches to a threshold, we then advance the curriculum by revealing one extra variable. Note that after revealing one extra variable, the training data now contains the observation of all revealed variables.

**Inference Results.** The test accuracy results are in Figure 2. Note that naive Bayes is able to model the first few variables in the selected graphs well as shown in the first column of Figure 2. However, as the order of the variable goes further, transformers outperforms naive Bayes on both sample efficiency and test accuracy. This indicates transformers are able to utilize the graph structure to generate prediction instead of treating all variables as independent observations. Notably, while our transformers are only trained on samples with $N_{\text{train}} = 100$, they are able to generalized to different values of $N_{\text{test}}$, and their test accuracy approaches to Bayesian inference when $N_{\text{test}}$ increases. This again verifies the capability of transformers to learn Bayesian inference and model graph structure well. Another thing to highlight is that both naive Bayes and Bayesian inference are not capable of handling unseen observations, leading to assigning 0 probability on every outcome under this case. However, transformers are able to utilize its learned prior from training data to perform prediction. This explains why transformers outperforms Bayesian inference sometimes when $N_{\text{test}}$ is small.

**Convergence Results.** We now discuss the convergence result of transformers training on general graph and tree in Figure 3. We show the loss and accuracy curve on training and test dataset throughout the optimization process. We also want to observe the generalization performance on $N$ of transformers. Specifically, we train models on $N = 100$, and evaluate them on both $N = 100$ and $N = 50$ cases. We observe that the loss curve presents a decreasing trend, and the accuracy is able to reach near optimal ($\sim 0.75$)[1] . Further, the generalization performance matches the results in Figure 2, as we see transformers are capable of performing Bayesian inference under different $N_{\text{test}}$.

## 5.2 GENERALIZATION ANALYSIS

Here we analyze when transformers trained on a fixed number of examples, which we denote $N_{\text{train}}$, whether it can generalize to different number of $N_{\text{test}}$. We evaluate 2 cases: (1) $N_{\text{train}} >> N_{\text{test}}$, (2) $N_{\text{train}} << N_{\text{test}}$. Note that in our construction, $N$ does not affect transformers ability to perform Bayesian inference. However, during training, small $N_{\text{train}}$ can produce large noise, whereas larger $N_{\text{train}}$, while being more stable, can be easily modeled by naive Bayes. This raises a doubt that whether transformers trained under larger $N_{\text{train}}$ learn naive Bayes or Bayesian inference. Therefore, we train transformers with $N_{\text{train}} \in \{5, 10, 200, 400\}$, and evaluate them with different $N_{\text{test}}$. We also

---

[1]This is a rouge estimation based on our design of probability distributions of training data.

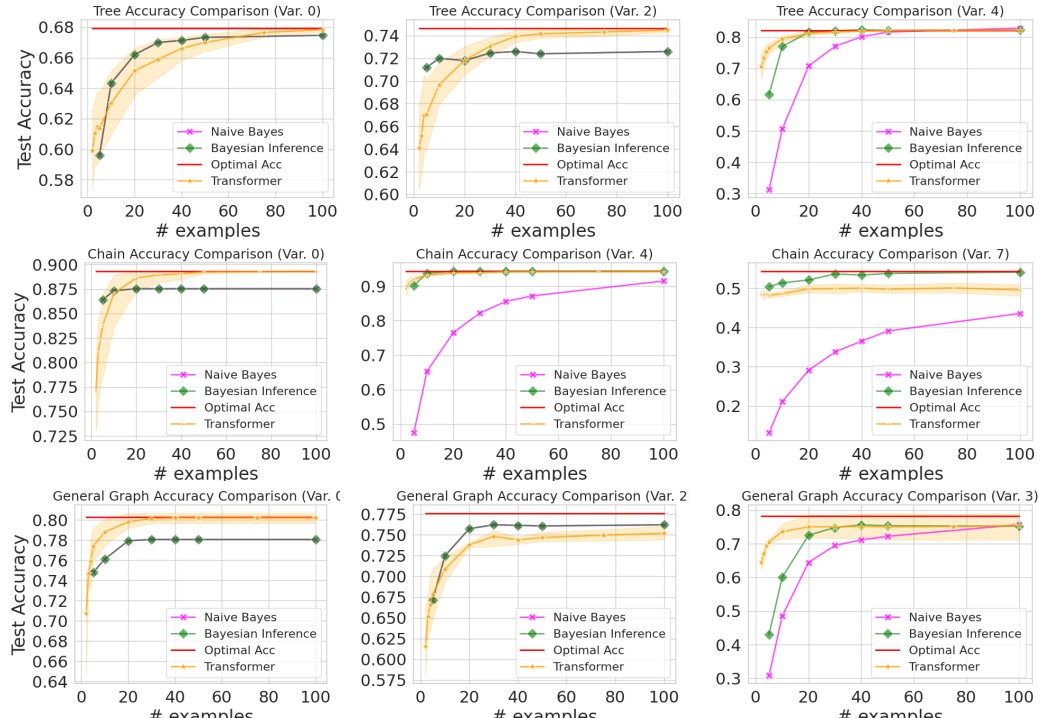

Figure 2: **Top to Bottom: The Accuracy Comparison on Tree, Chain and General Graphs.** We select 3 variables in each graph. For tree, we select one variable for each level from root to leaf. For chain, we select three variables that are close to the beginning, middle and the end of the chain. For graph, we select variables that are (1) no parents, (2) 2 parents, but the two parents have no precedents, and (3) 2 parents, and parents have other precedents. This setting makes (2) identical for naive Bayes and Bayesian inference, and (3) will present the difference. We observe transformers present similar performance with Bayesian inference and show better sample efficiency comparing to naive Bayes, indicating transformers are capable to model relationships between variables according to graph structure. Moreover, Bayesian inference and naive Bayes fail to generate prediction when the test token was never observed in the provided examples. However, transformers are able to generate predictions based on its learned prior, showing its superior performance under few examples.

report the loss and accuracy curve during training and use $N_{\text{test}} \in \{20, 50\}$ as testset. The choice of these numbers is based on the fact that these numbers are effective to show the gap between Bayesian inference and naive Bayes. We present the results on general graph in the main paper, the generalization analysis on tree can be found in Appendix B.3.

**Results.** The convergence and inference results are in Figure 4 and Figure 5, respectively. For the convergence result, we observe that models trained on large $N_{\text{train}}$ is able to generalize well on both $N_{\text{test}} = 20, 50$ (accuracy above 0.7). However, for models trained under small $N_{\text{train}}$, they do not converge well and also do not generalize well on testset (accuracy below 0.7). For the inference result, we see that models trained on large $N_{\text{train}}$ is capable of performing Bayesian inference. But models trained under small $N_{\text{train}}$ struggle to utilize the network structure to predict. A potential reason is smaller $N_{\text{train}}$ is not sufficient to approximate the ground truth probability distribution well. Also, while models trained on $N_{\text{train}} = 400$ is almost equivalent to learning on independent variables, modeling are still able to learn the network structure, potentially show the positive effect of curriculum. The result indicates that a sufficient large $N_{\text{train}}$ is critical for transformers to learn Bayesian inference in-context, providing practical insights on real-world scenarios and downstream tasks.

## 5.3 IS OUR CONSTRUCTION OPTIMAL?

We empirically evaluate whether our construction is optimal for transformers to learn Bayesian Inference. In our construction, a transformer needs at least 1 layer to perform Bayesian inference.

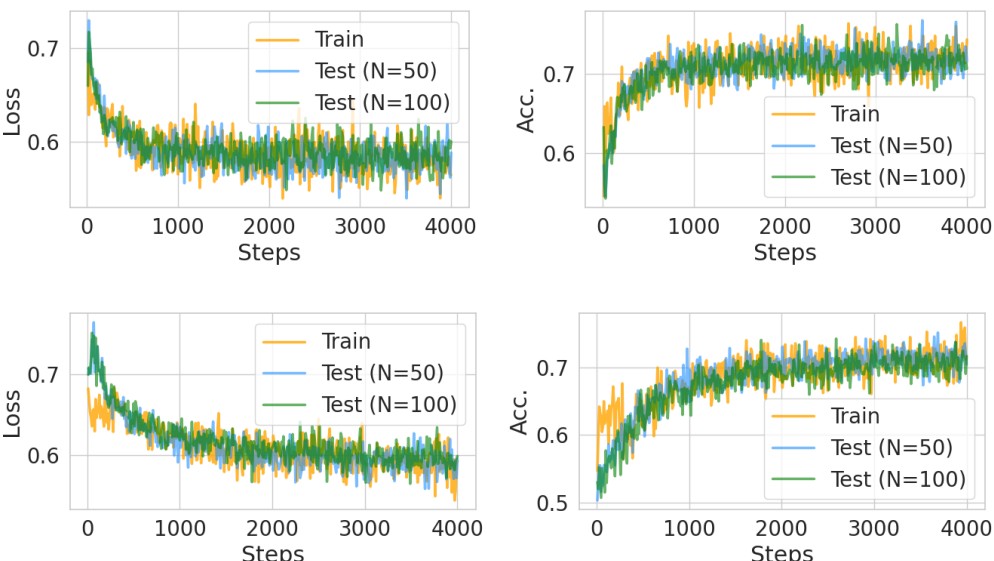

Figure 3: **Top: Convergence result on general graph. Bottom: Convergence result on tree.** We track the convergence result of transformers trained on general graph. Overall, we observe a decreasing trend of loss and increasing trend of accuracy on both training and test data. Note that all training and test samples are sampled from Bayesian networks. Therefore, the optimal loss and accuracy are not $0$ and $1$, respectively. We also see that transformers are able to generalize well on the $N_{\text{test}} = 50$ case even when its trained with $N_{\text{test}} = 100$. This shows transformers are capable of doing Bayesian inference instead of learning inductive bias of data. Note that during the beginning of training, transformers perform better on trainset than testset. This is due to the curriculum design that the model is initially exposed to only 2 variables, while evaluated on the whole graph.

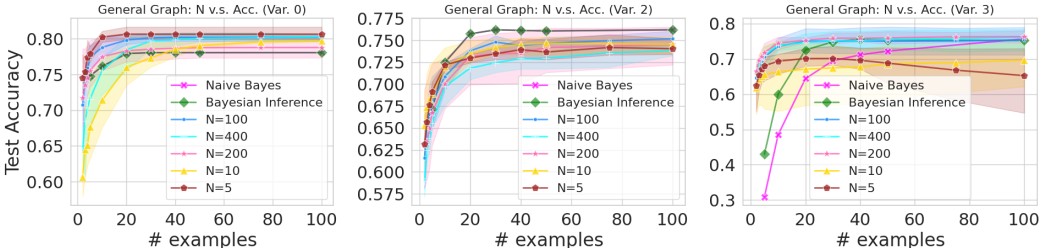

Figure 4: **Left to right: Transformer's performance on general graph variable 0, 2, 3.** For variable 0, 2, all models are able to model the variable distributions well. Interestingly, for variable 3, transformers trained under $N_{\text{train}} = [5, 10]$ are not capable of predicting it well. Moreover, its performance is even worse than naive Bayes for large $N_{\text{test}}$. The result indicates that a sufficient size of $N_{\text{train}}$ is necessary for transformers to learn the network structure.

To evaluate whether this requirement is necessary, we test transformers with different number of attention heads and different layers, to see if 1 layer are truly necessary for transformers to learn Bayesian inference.

**Results.** The results are in Figure 6. In general, transformers are capable of Bayesian inference empirically with either 1 layer, 1 head, or both. The performance difference between these variants are minimal. An intermediate conclusion is that even with 1-layer, 1-head, transformers are still capable of performing Bayesian inference on simple network structures. We interpret this result by making 2 assumptions: (1) The limitation of 1-layer or 1-head transformers on Bayesian inference is not presented by our general graph; (2) Our theoretical construction can be improved. For (1), it is possible that 1-layer, 1-head transformers are only capable of doing Bayesian inference on simple

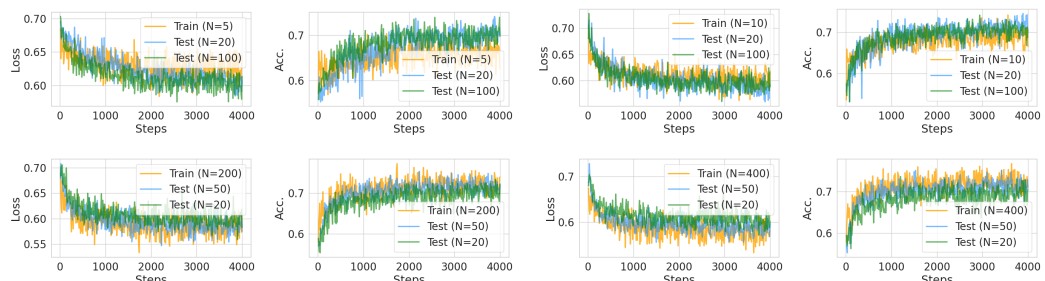

Figure 5: **Left: Convergence result on general graph for $N_{\text{train}} \in \{5, 10, 200, 400\}$. Right: Convergence result on tree for $N_{\text{train}} \in \{5, 10, 200, 400\}$.** Here we observe an obvious contrast between models trained on large and small $N_{\text{train}}$. For smaller $N_{\text{train}}$, model performance on training dataset is lower than testset. For larger $N_{\text{train}}$, we observe the opposite. We believe this is due to the fact that smaller $N_{\text{train}}$ does not provide sufficient sample size to recover the probability distribution well.

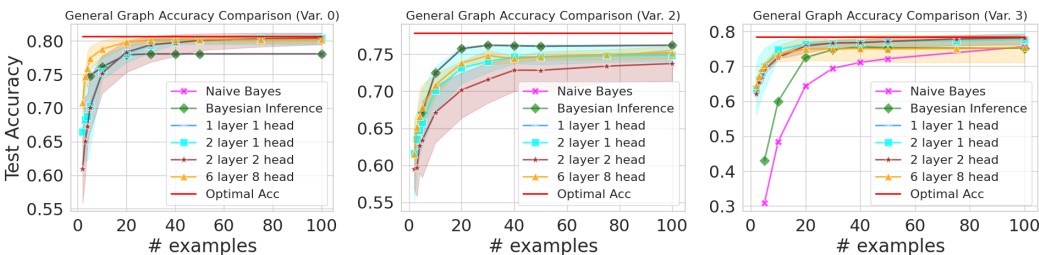

Figure 6: **Accuracy comparison on general graph between transformers with different hyper-parameters. Left to right: evaluation on variable 0, 2, 3.** In general, we found out that with low number of layers or attention heads, transformers still perform similarly with the 6-layer counterpart. Especially with the 1-layer 1-head version, it is still capable of performing Bayesian inference and presents only slight performance drop.

network structures but not on networks with complex structure such as large number of parents, sparse connection etc. A possibility is that such a network structure bottleneck for 1-layer, 1-head exists but not presented in our general graph. For (2), we also hypothesize that our construction might not utilize the full potential of attention layers or the feed-forward layers. For the further analysis of these assumptions, we leave for future work.

## 6 PROOF SKETCH

In this section, we give a proof sketch of Theorem 4.1. Te proof is based on relatively intuitive constructions of the two transformer layers. The result for the first transformer layer is summarized into the following lemma.

**Lemma 6.1.** For any Bayesian network $\mathcal{B}$ with maximum in-degree $D$, there exists a one-layer transformer $\text{TF}_{\boldsymbol{\theta}^{(1)}}(\cdot)$ with parameter matrices satisfying $\|\mathbf{V}^{(1)}\|_2, \|\mathbf{K}^{(1)}\|_2, \|\mathbf{Q}^{(1)}\|_2, \|\mathbf{W}_2^{(1)}\|_2 \leq 1$ and $\|\mathbf{W}_1^{(1)}\|_2 \leq 2\sqrt{D+1}$, such that for any the variable-of-interest index $m_0$, it holds that

$$
\text{TF}_{\boldsymbol{\theta}^{(1)}}(\mathbf{X}) = \widetilde{\mathbf{X}} := \begin{bmatrix} \widetilde{\mathbf{x}}_{11} & \widetilde{\mathbf{x}}_{12} & \cdots & \widetilde{\mathbf{x}}_{1N} & \widetilde{\mathbf{x}}_{1q} \\ \widetilde{\mathbf{x}}_{21} & \widetilde{\mathbf{x}}_{22} & \cdots & \widetilde{\mathbf{x}}_{2N} & \widetilde{\mathbf{x}}_{2q} \\ \vdots & \vdots & & \vdots & \vdots \\ \widetilde{\mathbf{x}}_{M1} & \widetilde{\mathbf{x}}_{M2} & \cdots & \widetilde{\mathbf{x}}_{MN} & \widetilde{\mathbf{x}}_{Mq} \\ \mathbf{p} & \mathbf{p} & \cdots & \mathbf{p} & \mathbf{p}_q \end{bmatrix},
$$

where

$$
\widetilde{\mathbf{x}}_{mi} = \begin{cases} \mathbf{x}_{mi}, & \text{if } m \in \{m_0\} \cup \mathcal{P}(m_0); \\ \mathbf{0}, & \text{otherwise.} \end{cases}, \quad \widetilde{\mathbf{x}}_{mq} = \begin{cases} \mathbf{x}_{mq}, & \text{if } m \in \{m_0\} \cup \mathcal{P}(m_0); \\ \mathbf{0}, & \text{otherwise.} \end{cases}
$$

for all $i \in [N]$.

Lemma 6.1 above shows that, there exists a transformer layer with bounded weight matrices that can serves as a "parents selector" – for any $m_0 \in [M]$, as long as the "positional embeddings" $\mathbf{p}$ and $\mathbf{p}_q$ are defined accordingly, the output of the transformer layer will retain only the values of the observed variables that are direct parents of the $m_0$-th variable. This operation, which trims all non-essential observation values, effectively prepares for the in-context estimation of the conditional probabilities in the second layer.

The following lemma gives the result for the second transformer layer, which takes the output $\widetilde{\mathbf{X}}$ of the first layer given in Lemma 6.1 as input.

**Lemma 6.2.** For any $\epsilon > 0$ and any Bayesian network $\mathcal{B}$ with maximum in-degree $D$, there exists a one-layer transformer $\mathrm{TF}_{\boldsymbol{\theta}^{(2)}}(\cdot)$ with parameter matrices satisfying $\|\mathbf{V}^{(2)}\|_2, \|\mathbf{W}_1^{(2)}\|_2, \|\mathbf{W}_2^{(2)}\|_2 \leq 1$ and $\|\mathbf{K}^{(2)}\|_2, \|\mathbf{Q}^{(2)}\|_2 \leq 3\log(MdN/\epsilon)$, such that for any index of the variable-of-interest $m_0$ and the corresponding $\widetilde{\mathbf{X}}$ defined in Lemma 6.1, it holds that

$$\mathrm{Read}\big[\mathrm{TF}_{\boldsymbol{\theta}^{(2)}}(\widetilde{\mathbf{X}})\big] = \widehat{\mathbf{x}}_q + \mathbf{s},$$

where $\widehat{\mathbf{x}}_q = [\mathbf{0}_{(m_0-1)d}^\top, -\widehat{\mathbf{x}}_{m_0q}^\top, \mathbf{0}_{(2M-m_0)d}^\top, \mathbf{1}_d^\top]^\top$ with $\widehat{\mathbf{x}}_{m_0q} = \mathbf{p}_{m_0}^{\mathrm{opt}}$ and $\|\mathbf{s}\|_\infty \leq \epsilon/[(2M+1)d]$.

Lemma 6.2 shows that, there exists a transformer layer which takes the output of $\widetilde{\mathbf{X}}$ defined in Lemma 6.1 as input, and outputs a matrix whose last column is directly related to the target optimal maximum likelihood estimation $\mathbf{p}_{m_0}^{\mathrm{opt}}$.

Given the two lemmas above, the proof of Theorem 4.1 is straightforward. The proof is as follows.

*Proof of Theorem 4.1.* Let $\mathrm{TF}_{\boldsymbol{\theta}^{(1)}}(\cdot)$ and $\mathrm{TF}_{\boldsymbol{\theta}^{(2)}}(\cdot)$ be defined in Lemmas 6.1 and 6.2 respectively. Then we directly have

$$\mathrm{Read}\big[\mathrm{TF}_{\boldsymbol{\theta}^{(2)}}(\mathrm{TF}_{\boldsymbol{\theta}^{(1)}}(\mathbf{X}))\big] = \widehat{\mathbf{x}}_q + \mathbf{s},$$

where $\widehat{\mathbf{x}}_q = [\mathbf{0}_{(m_0-1)d}^\top, -\widehat{\mathbf{x}}_{m_0q}^\top, \mathbf{0}_{(2M-m_0)d}^\top, \mathbf{1}_d^\top]^\top$ with $\widehat{\mathbf{x}}_{m_0q} = \mathbf{p}_{m_0}^{\mathrm{opt}}$ and $\|\mathbf{s}\|_\infty \leq \epsilon/[(2M+1)d]$. Therefore, setting $\mathbf{A} = [\mathbf{0}_{d\times(m_0-1)d}, -\mathbf{I}_{d\times d}, \mathbf{0}_{d\times(2M-m_0+1)d}]$, we obtain

$$\mathbf{A}\,\mathrm{Read}\big[\mathrm{TF}_{\boldsymbol{\theta}^{(2)}}(\mathrm{TF}_{\boldsymbol{\theta}^{(1)}}(\mathbf{X}))\big] = \mathbf{A}\widehat{\mathbf{x}}_q + \mathbf{As} = \mathbf{p}_{m_0}^{\mathrm{opt}} + \mathbf{As}.$$

By definition, it is clear that $\|\mathbf{As}\|_\infty \leq \epsilon/d$. This finishes the proof. $\qquad\square$

# 7 CONCLUSION

In this paper, we theoretically analyze transformer's capability to learn Bayesian networks in-context in an autoregressive fashion. We show that there exists a simple construction of transformer such that it can (1) estimate the conditional probabilities of the Bayesian network in-context, and (2) autoregressively generate a new sample based on the estimated conditional probabilities. This sheds light on the potential of transformers in probabilistic reasoning and their applicability in various machine learning tasks involving structured data. Empirically, we provide extensive experiments to show that transformers are indeed capable of learning Bayesian networks and generalize well on unseen probability distributions, verifying our theoretical construction. Our theoretical and experimental results provide not only greater insights on the understanding of transformers, but also practical guidance in training transformers on Bayesian networks.

There are still multiple important aspects which this paper does not cover. First of all, our current theoretical result only demonstrates the *expressive power* of transformers in the sense that a good transformer model with reasonable weights exist. Our result does not directly cover whether such a transformer can indeed be obtained through training. Our experiments indicate a positive answer to this question, making theoretical demonstrations a promising future work direction. Moreover, our current analysis does not take the number of heads into consideration. As is discussed in Nichani et al. (2024), multi-head attention may play an important role when learning Bayesian networks with complicated network structures. Studying the impact of multi-head attention is another important future work direction.

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

## A PROOFS

In this section, we give the proofs of Lemmas 6.1 nad 6.2.

### A.1 PROOF OF LEMMA 6.1

The proof of Lemma 6.1 is given as follows.

*Proof of Lemma 6.1.* Let $\mathbf{V}^{(1)} = \mathbf{0}_{(2M+1)d \times (2M+1)d}$, $\mathbf{K}^{(1)} = \mathbf{Q}^{(1)} = \mathbf{0}_{Md \times (2M+1)}$. Then clearly we have

$$\text{Attn}_{\mathbf{V}^{(1)}, \mathbf{K}^{(1)}, \mathbf{Q}^{(1)}}(\mathbf{X}) = \mathbf{X}.$$

Moreover, let $\mathbf{A} = [\mathbf{A}_{ij}]_{M \times (M+2)} \in \mathbb{R}^{Md \times (M+1)d}$ be a $M \times (M+1)$ block matrix where

$$\mathbf{A}_{ij} = \begin{cases} \mathbf{I}_{d \times d}, & \text{if } j \leq M \text{ and } i \in \{j\} \cup \mathcal{P}(j); \\ \mathbf{0}_{d \times d}, & \text{otherwise.} \end{cases}$$

Then, let $\mathbf{W}_2^{(1)} = -\mathbf{I}_{(2M+1)d \times (2M+1)d}$, and

$$\mathbf{W}_1^{(1)} = \begin{bmatrix} \mathbf{I}_{Md \times Md} & -2\mathbf{A} \\ \mathbf{0}_{(M+1)d \times Md} & \mathbf{0}_{(M+1)d \times (M+1)d} \end{bmatrix}.$$

We note that the above definintion does not rely on any specific value of $m_0$. By definition, we can directly verify that

$$\mathbf{W}_1^{(1)} \mathbf{X} = \begin{bmatrix} \check{\mathbf{x}}_{11} & \check{\mathbf{x}}_{12} & \cdots & \check{\mathbf{x}}_{1N} & \check{\mathbf{x}}_{1q} \\ \check{\mathbf{x}}_{21} & \check{\mathbf{x}}_{22} & \cdots & \check{\mathbf{x}}_{2N} & \check{\mathbf{x}}_{2q} \\ \vdots & \vdots & & \vdots & \vdots \\ \check{\mathbf{x}}_{M1} & \check{\mathbf{x}}_{M2} & \cdots & \check{\mathbf{x}}_{MN} & \check{\mathbf{x}}_{Mq} \\ \mathbf{0}_{(M+1)d} & \mathbf{0}_{(M+1)d} & \cdots & \mathbf{0}_{(M+1)d} & \mathbf{0}_{(M+1)d} \end{bmatrix},$$

where $\check{\mathbf{x}}_{mi} = \mathbf{x}_{mi} - 2\mathbf{1} \cdot \mathbb{1}[m \in \{m_0\} \cup \mathcal{P}(m_0)]$. Now since $\mathbf{x}_{mi}$, $m \in [M]$, $i \in [N]$ are all one-hot vectors (and therefore have non-negative entries between zero and one), we see that the entries of $\check{\mathbf{x}}_{mi}$ are strictly negative if and only if $m \in \{m_0\} \cup \mathcal{P}(m_0)$. Therefore, by the definition of the

ReLU activation function, we have

$$\sigma(\mathbf{W}_1^{(1)}\mathbf{X}) = \begin{bmatrix} \overline{\mathbf{x}}_{11} & \overline{\mathbf{x}}_{12} & \cdots & \overline{\mathbf{x}}_{1N} & \overline{\mathbf{x}}_{1q} \\ \overline{\mathbf{x}}_{21} & \overline{\mathbf{x}}_{22} & \cdots & \overline{\mathbf{x}}_{2N} & \overline{\mathbf{x}}_{2q} \\ \vdots & \vdots & & \vdots & \vdots \\ \overline{\mathbf{x}}_{M1} & \overline{\mathbf{x}}_{M2} & \cdots & \overline{\mathbf{x}}_{MN} & \overline{\mathbf{x}}_{Mq} \\ \mathbf{0}_{(M+1)d} & \mathbf{0}_{(M+1)d} & \cdots & \mathbf{0}_{(M+1)d} & \mathbf{0}_{(M+1)d} \end{bmatrix},$$

where $\overline{\mathbf{x}}_{mi} = \mathbf{x}_{mi} \cdot \mathbb{1}[m \notin \{m_0\} \cup \mathcal{P}(m_0)]$. Therefore, by $\mathbf{W}_2^{(1)} = -\mathbf{I}_{2Md \times 2Md}$, we have

$$\mathrm{TF}_{\boldsymbol{\theta}^{(1)}}(\mathbf{X}) = \mathrm{FF}_{\mathbf{W}_1^{(1)}, \mathbf{W}_2^{(1)}}[\mathrm{Attn}_{\mathbf{V}^{(1)}, \mathbf{K}^{(1)}, \mathbf{Q}^{(1)}}(\mathbf{X})] = \mathrm{FF}_{\mathbf{W}_1^{(1)}, \mathbf{W}_2^{(1)}}(\mathbf{X})$$

$$= \mathbf{X} + \mathbf{W}_2^{(1)}\sigma(\mathbf{W}_1^{(1)}\mathbf{X}) = \mathbf{X} - \sigma(\mathbf{W}_1^{(1)}\mathbf{X})$$

$$= \begin{bmatrix} \widetilde{\mathbf{x}}_{11} & \widetilde{\mathbf{x}}_{12} & \cdots & \widetilde{\mathbf{x}}_{1N} & \widetilde{\mathbf{x}}_{1q} \\ \widetilde{\mathbf{x}}_{21} & \widetilde{\mathbf{x}}_{22} & \cdots & \widetilde{\mathbf{x}}_{2N} & \widetilde{\mathbf{x}}_{2q} \\ \vdots & \vdots & & \vdots & \vdots \\ \widetilde{\mathbf{x}}_{M1} & \widetilde{\mathbf{x}}_{M2} & \cdots & \widetilde{\mathbf{x}}_{MN} & \widetilde{\mathbf{x}}_{Mq} \\ \mathbf{p} & \mathbf{p} & \cdots & \mathbf{p} & \mathbf{p}_q \end{bmatrix},$$

where

$$\widetilde{\mathbf{x}}_{mi} = \begin{cases} \mathbf{x}_{mi}, & \text{if } m \in \{m_0\} \cup \mathcal{P}(m_0); \\ \mathbf{0}, & \text{otherwise.} \end{cases}, \quad \widetilde{\mathbf{x}}_{mq} = \begin{cases} \mathbf{x}_{mq}, & \text{if } m \in \{m_0\} \cup \mathcal{P}(m_0); \\ \mathbf{0}, & \text{otherwise.} \end{cases}$$

for all $i \in [N]$. This finishes the proof. $\qquad \square$

## A.2 PROOF OF LEMMA 6.2

We present the proof of Lemma 6.2 as follows.

*Proof of Lemma 6.2.* Clearly, by the definition of the $\mathrm{Read}(\cdot)$ function, only the last column of the output of $\mathrm{TF}_{\boldsymbol{\theta}^{(2)}}$ matters. Since the last column of the output of $\mathrm{TF}_{\boldsymbol{\theta}^{(2)}}$ only relies on the last column of $\mathrm{Attn}_{\mathbf{V}^{(2)}, \mathbf{K}^{(2)}, \mathbf{Q}^{(2)}}(\widetilde{\mathbf{X}})$, we focus on the last column of $\mathrm{softmax}[(\mathbf{KX})^\top(\mathbf{QX})]$, which is $\mathrm{softmax}[(\mathbf{K}\widetilde{\mathbf{X}})^\top(\mathbf{Q}\widetilde{\mathbf{x}}_q)]$, where $\widetilde{\mathbf{x}}_q = [\widetilde{\mathbf{x}}_{1q}^\top, \ldots, \widetilde{\mathbf{x}}_{Mq}^\top, \mathbf{p}_q^\top]^\top$. Denote $c = \log(d/\epsilon)$. Let $\mathbf{W}_1^{(2)} = \mathbf{W}_2^{(2)} = \mathbf{0}_{(2M+1)d \times (2M+1)d}$, $\mathbf{V}^{(2)} = -\mathbf{I}_{(2M+1)d \times (2M+1)d}$, and

$$\mathbf{K}^{(2)} = \sqrt{c} \cdot \begin{bmatrix} \mathbf{I}_{Md \times Md} & \mathbf{0}_{Md \times Md} & \mathbf{0}_{Md \times d} \\ \mathbf{0}_{d \times Md} & \mathbf{0}_{d \times Md} & \mathbf{I}_{d \times d} \end{bmatrix}, \quad \mathbf{Q}^{(2)} = \sqrt{c} \cdot \begin{bmatrix} \mathbf{I}_{Md \times Md} & \mathbf{0}_{Md \times Md} & \mathbf{0}_{Md \times d} \\ \mathbf{0}_{d \times Md} & \mathbf{0}_{d \times Md} & -\mathbf{I}_{d \times d} \end{bmatrix}.$$

Then we have

$$\mathbf{K}^{(2)}\widetilde{\mathbf{X}} = \sqrt{c} \cdot \begin{bmatrix} \widetilde{\mathbf{x}}_{11} & \widetilde{\mathbf{x}}_{12} & \cdots & \widetilde{\mathbf{x}}_{1N} & \widetilde{\mathbf{x}}_{1q} \\ \widetilde{\mathbf{x}}_{21} & \widetilde{\mathbf{x}}_{22} & \cdots & \widetilde{\mathbf{x}}_{2N} & \widetilde{\mathbf{x}}_{2q} \\ \vdots & \vdots & & \vdots & \vdots \\ \widetilde{\mathbf{x}}_{M1} & \widetilde{\mathbf{x}}_{M2} & \cdots & \widetilde{\mathbf{x}}_{MN} & \widetilde{\mathbf{x}}_{Mq} \\ \mathbf{0}_d & \mathbf{0}_d & \cdots & \mathbf{0}_d & \mathbf{1}_d \end{bmatrix}, \quad \mathbf{Q}^{(2)}\widetilde{\mathbf{x}}_q = \sqrt{c} \cdot \begin{bmatrix} \widetilde{\mathbf{x}}_{1q} \\ \widetilde{\mathbf{x}}_{2q} \\ \vdots \\ \widetilde{\mathbf{x}}_{Mq} \\ -\mathbf{1}_d \end{bmatrix}.$$

Recall the definition that

$$\widetilde{\mathbf{x}}_{mi} = \begin{cases} \mathbf{x}_{mi}, & \text{if } m \in \{m_0\} \cup \mathcal{P}(m_0); \\ \mathbf{0}, & \text{otherwise.} \end{cases}, \quad \widetilde{\mathbf{x}}_{mq} = \begin{cases} \mathbf{x}_{mq}, & \text{if } m \in \{m_0\} \cup \mathcal{P}(m_0); \\ \mathbf{0}, & \text{otherwise.} \end{cases}$$

for all $i \in [N]$. Therefore, for $i \in [N]$, we have

$$(\mathbf{K}\widetilde{\mathbf{x}}_i)^\top(\mathbf{Q}\widetilde{\mathbf{x}}_q) = c \cdot \sum_{m=1}^{M} \langle \widetilde{\mathbf{x}}_{mi}, \widetilde{\mathbf{x}}_{mq} \rangle$$

$$= c \cdot \sum_{m=1}^{M} \langle \mathbf{x}_{mi}, \mathbf{x}_{mq} \rangle \mathbb{1}[m \in \{m_0\} \cup \mathcal{P}(m_0)]$$

$$= c \cdot |\{m \in \{m_0\} \cup \mathcal{P}(m_0) : \mathbf{x}_{mi} = \mathbf{x}_{mq}\}|$$

$$= c \cdot |\{m \in \mathcal{P}(m_0) : \mathbf{x}_{mi} = \mathbf{x}_{mq}\}|,$$

where the last equation is due to the fact that $\mathbf{x}_{m_0q} = \mathbf{0}$, as it has not been sampled. Similarly, we also have

$$(\mathbf{K}\widetilde{\mathbf{x}}_q)^\top(\mathbf{Q}\widetilde{\mathbf{x}}_q) = c \cdot \sum_{m=1}^{M} \langle \widetilde{\mathbf{x}}_{mq}, \widetilde{\mathbf{x}}_{mq} \rangle - cd = c \cdot |\mathcal{P}(m_0)| - cd.$$

Now denote $\mathcal{I}(m_0) = \{i \in [N] : \mathbf{x}_{mi} = \mathbf{x}_{mq} \text{ for all } m \in \mathcal{P}(m_0)\}$. Then for any $i \in \mathcal{I}(m_0)$ (by assumption, this set is not empty), we have

$$|\{m \in \mathcal{P}(m_0) : \mathbf{x}_{mi} = \mathbf{x}_{mq}\}| = |\mathcal{P}(m_0)|.$$

Therefore, for any $i \in \mathcal{I}(m_0)$ and any $i' \notin \mathcal{I}(m_0)$, we have

$$(\mathbf{K}\widetilde{\mathbf{x}}_i)^\top(\mathbf{Q}\widetilde{\mathbf{x}}_q) - (\mathbf{K}\widetilde{\mathbf{x}}_{i'})^\top(\mathbf{Q}\widetilde{\mathbf{x}}_q) \geq c \cdot |\mathcal{P}(m_0)| - c \cdot (|\mathcal{P}(m_0)| - 1) = c.$$

Moreover,

$$(\mathbf{K}\widetilde{\mathbf{x}}_i)^\top(\mathbf{Q}\widetilde{\mathbf{x}}_q) - (\mathbf{K}\widetilde{\mathbf{x}}_q)^\top(\mathbf{Q}\widetilde{\mathbf{x}}_q) = c \cdot |\mathcal{P}(m_0)| - c \cdot |\mathcal{P}(m_0)| + cd = cd.$$

Therefore, by $c = 3\log(MdN/\epsilon)$ we have

$$\left\| \text{softmax}[(\mathbf{K}\widetilde{\mathbf{X}})^\top(\mathbf{Q}\widetilde{\mathbf{x}}_q)] - \frac{1}{|\mathcal{I}(m_0)|} \sum_{i \in \mathcal{I}(m_0)} \mathbf{e}_i \right\|_\infty \leq \frac{\epsilon}{(2M+1)d}.$$

Now by the choice that $-\mathbf{V}^{(2)} = \mathbf{I}_{(2M+1)d \times (2M+1)d}$, we have

$$\text{Read}\big[\text{Attn}_{\mathbf{V}^{(2)}, \mathbf{K}^{(2)}, \mathbf{Q}^{(2)}}(\widetilde{\mathbf{X}})\big] = \widetilde{\mathbf{x}}_q + \mathbf{V}^{(2)}\widetilde{\mathbf{X}}\text{softmax}[(\mathbf{K}^{(2)}\widetilde{\mathbf{X}})^\top(\mathbf{Q}^{(2)}\widetilde{\mathbf{x}}_q)]$$

$$= \widetilde{\mathbf{x}}_q - \frac{1}{|\mathcal{I}(m_0)|} \sum_{i \in \mathcal{I}(m_0)} \widetilde{\mathbf{X}}\mathbf{e}_i + \mathbf{s}$$

where $\mathbf{s} \in \mathbb{R}^{(2M+1)d}$ satisfies $\|\mathbf{s}\|_\infty \leq \epsilon/[(2M+1)d]$. Now note that (i) $\widetilde{\mathbf{x}}_{mi}$'s and $\widetilde{\mathbf{x}}_{mq}$'s are all zero except for $m \in \{m_0\} \cup \mathcal{P}(m_0)$, (ii) for all $i \in \mathcal{I}(m_0)$, and $m \in \mathcal{P}(m_0)$, $\mathbf{x}_{mi} = \mathbf{x}_{mq}$. Therefore, on the right-hand side of the equation above, most of the terms are actually canceled when calculating the difference $\widetilde{\mathbf{x}}_q - \frac{1}{|\mathcal{I}(m_0)|} \sum_{i \in \mathcal{I}(m_0)} \widetilde{\mathbf{X}}\mathbf{e}_i$. We have

$$\text{Read}\big[\text{Attn}_{\mathbf{V}^{(2)}, \mathbf{K}^{(2)}, \mathbf{Q}^{(2)}}(\widetilde{\mathbf{X}})\big] = \widehat{\mathbf{x}}_q + \mathbf{s},$$

where $\widehat{\mathbf{x}}_q = [\mathbf{0}_{(m_0-1)d}^\top, -\widehat{\mathbf{x}}_{m_0q}^\top, \mathbf{0}_{(2M-m_0)d}^\top, \mathbf{1}_d^\top]^\top$, and

$$\widehat{\mathbf{x}}_{m_0q} = \frac{1}{|\mathcal{I}(m_0)|} \sum_{i \in \mathcal{I}(m_0)} \widetilde{\mathbf{x}}_{m_0i}$$

$$= \frac{1}{|\mathcal{I}(m_0)|} \sum_{i \in \mathcal{I}(m_0)} \mathbf{x}_{m_0i}$$

$$= \sum_{i \in [N]} \mathbf{x}_{m_0i} \frac{\mathbb{1}[\mathbf{x}_{mi} = \mathbf{x}_{mq} \text{ for all } m \in \mathcal{P}(m_0)]}{|\{i \in [N] : \mathbf{x}_{mi} = \mathbf{x}_{mq} \text{ for all } m \in \mathcal{P}(m_0)\}|}.$$

Now by $\mathbf{W}_1^{(2)} = \mathbf{0}_{(2M+1)d \times (2M+1)d}$, $\mathbf{W}_2^{(2)} = \mathbf{0}_{(2M+1)d \times (2M+1)d}$, we have

$$\text{Read}\big[\text{TF}_{\boldsymbol{\theta}^{(2)}}(\widetilde{\mathbf{X}})\big] = \text{Read}\big[\text{FF}_{\mathbf{W}_1^{(2)}, \mathbf{W}_2^{(2)}}[\text{Attn}_{\mathbf{V}^{(2)}, \mathbf{K}^{(2)}, \mathbf{Q}^{(2)}}(\widetilde{\mathbf{X}})]\big]$$

$$= \text{FF}_{\mathbf{W}_1^{(2)}, \mathbf{W}_2^{(2)}}\big\{\text{Read}\big[\text{Attn}_{\mathbf{V}^{(2)}, \mathbf{K}^{(2)}, \mathbf{Q}^{(2)}}(\widetilde{\mathbf{X}})\big]\big\}$$

$$= \text{FF}_{\mathbf{W}_1^{(2)}, \mathbf{W}_2^{(2)}}(\widehat{\mathbf{x}}_q + \mathbf{s})$$

$$= \widehat{\mathbf{x}}_q + \mathbf{s}.$$

This finishes the proof. $\qquad\square$

# B   ADDITIONAL EXPERIMENTS

Here we conduct a hyperparameter analysis to see whether transformers are sensitive on certain hyperparameters. It is also a more complete result of some experimental sections in main paper. We analyze three hyperparameters:

- Number of layers
- Number of attention heads
- $N_{\text{train}}$

We perform these analysis on general graph and select variable 0, 2, 3 to evaluate. The reasoning behind this selection is to demonstrate 3 different properties of these variables. For variable 0, it is a random variable without any parents, so modeling it is

## B.1   THE EFFECT OF LAYERS.

Here we evaluate transformers with $\{1, 2, 6\}$ layers on general graph. Overall, we want to observe whether the number of layers affect transformer's ability to learn Bayesian inference. The result is in Figure 7.

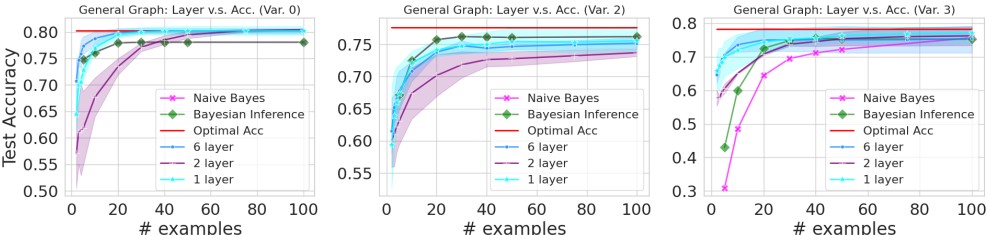

Figure 7: **Evaluation of transformers with different layer on general graph. Left to right: variable 0, 2, 3.** We set the hidden dimension to 256, number of heads to 8 for all transformers. The result is the average taken over 5 runs. We observe that even the 2-layer transformer performs worse and presents larger variance, all transformers have similar behavior on this task.

## B.2   THE EFFECT OF HEADS.

Here we evaluate transformers with $\{1, 2, 4, 8\}$ attention heads on general graph. Overall, we want to observe whether the number of attention heads affect transformer's ability to learn Bayesian inference. The result is in Figure 8. Empirically, we do not discover a significant impact of attention heads on models performance in our case study. As discussed in section 5.3, while we do not observe such an impact, it might due to the fact that the general graph structure is too simple to reflect such an architecture bottleneck.

## B.3   THE EFFECT OF $N$ DURING TRAINING.

Here we evaluate transformers with values of $N_{\text{train}}$ on general graph and tree. We aim to test models generalization capability and evaluate whether models require certain size of $N_{\text{train}}$ to learn Bayesian inference in-context.

**General Graph.**   The convergence and inference results are in Figure 10 and Figure 9, respectively. For the convergence result, we observe that models trained on large $N_{\text{train}}$ is able to generalize well on both $N_{\text{test}} = 20, 50$ (accuracy above 0.7). However, for models trained under small $N_{\text{train}}$, they do not converge well and also do not generalize well on testset (accuracy below 0.7). For the inference result, we see that models trained on large $N_{\text{train}}$ is capable of performing Bayesian inference. But models trained under small $N_{\text{train}}$ struggle to utilize the network structure to predict. A potential reason is smaller $N_{\text{train}}$ is not sufficient to approximate the ground truth probability distribution well. The result indicates that a sufficient large $N_{\text{train}}$ is critical for transformers to learn Bayesian inference in-context, providing practical insights on real-world scenarios and downstream tasks.

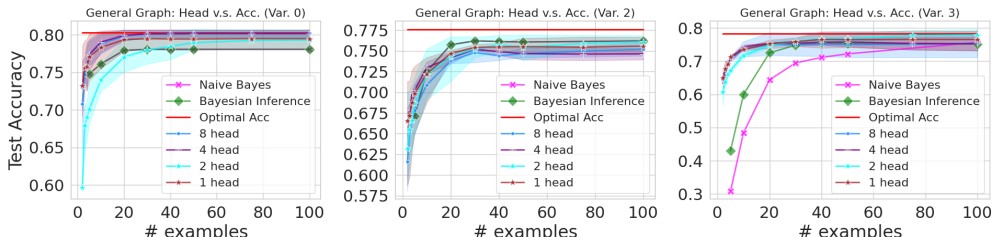

Figure 8: **Evaluation of transformers with different number of attention heads on general graph. Left to right: variable 0, 2, 3.** We set the hidden dimension to $256$, layer to $6$ for all transformers. The result is the average taken over 5 runs. Similar to the above subsection, we also do not observe significant performance degradation when reducing the number of heads. Especially for variable 3, which highly requires the network structure to inference prediction, transformer with 1-head still performs similar with its other variants.

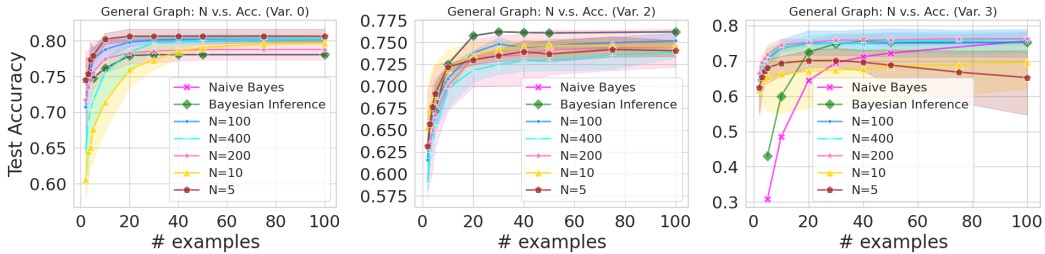

Figure 9: **Left to right: Transformer's performance on general graph variable 0, 2, 3.** For variable 0, 2, all models are able to model the variable distributions well. Interestingly, for variable 3, transformers trained under $N_{\text{train}} = [5, 10]$ are not capable of predicting it well. Moreover, its performance even worse than naive Bayes for large $N_{\text{test}}$. The result indicates that a sufficient size of $N_{\text{train}}$ is necessary for transformers to learn the network structure.

**Tree.** The results are demonstrated in Figure 11 and Figure 12. Overall, we observe that transformers fail to perform Bayesian inference when $N_{\text{train}} = 5$. However, different from our results on general graph, $N_{\text{train}} = 10$ seems to be sufficient for transformers to learn Bayesian inference. This result can be explained by the fact that modeling variable 4 only requires to focus on its single parent. However, in general graphs, some variables have multiple parents, which prevents $N_{\text{train}} = 10$ to recover the conditional probability distribution well.

### B.4 ADDITIONAL EXPERIMENT FOR CATEGORICAL DISTRIBUTIONS

Here we conduct experiments on networks with categorical distributions, i.e. the number of possible outcome for each variable is more than 2. We select the binary tree structure as example, and set the number of possible outcome for each variable as 3. We report both the test accuracy and test F1 are evaluation metrics, the results are in Figure 13 and Figure 14. As a result, the input dimension of the transformer is 28. For all other hyperparameters, we following Table 1.

## C EXPERIMENTAL DETAILS

### C.1 DATA DETAILS

Here we provide visualizations of graphs structures we select in our experiments. Arrows indicates the causal relationship between variables. Specifically, the "general graph" contains variables with more than 1 parents, representing a more generalized case. An interesting design of the general graph is its variable 2 and 3 are both governed by 2 parents. However, modeling variable 2 can be done via naive Bayes while modeling variable 3 requires Bayesian inference, giving us an opportunity to discover such property.

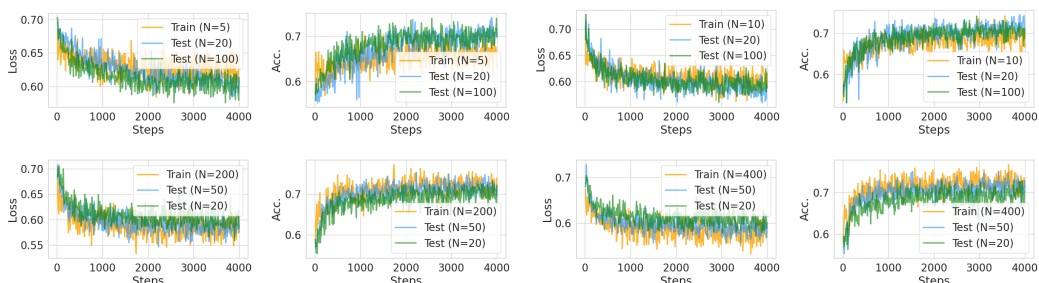

Figure 10: **Left: Convergence result on general graph for** $N_{\text{train}} \in \{5, 10, 200, 400\}$**. Right: Convergence result on tree for** $N_{\text{train}} \in \{5, 10, 200, 400\}$**.** Here we observe an obvious contrast between models trained on different $N_{\text{train}}$. For smaller $N_{\text{train}}$, model performance on training dataset is lower than testset. For larger $N_{\text{train}}$, we observe the opposite. We believe this is due to the fact that smaller $N_{\text{train}}$ does not provide sufficient sample size to recover the probability distribution well.

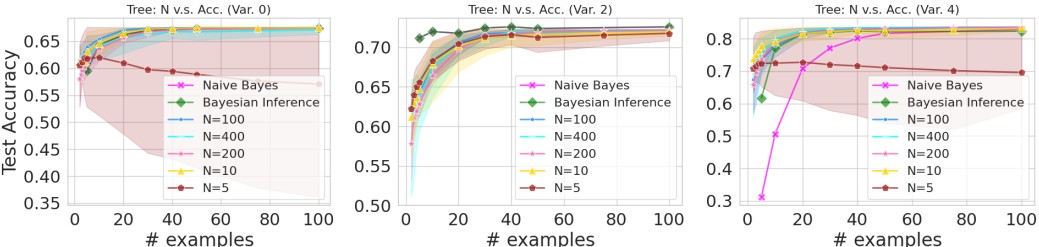

Figure 11: **Generalization Analysis: Inference results on tree.** Similar to our results on graph, transformers trained on large $N_{\text{train}}$ generalize better than trained on smaller $N_{\text{train}}$. Especially with $N_{\text{train}} = 5$, transformers fail to even predict well

## C.2 TRAINING DETAILS

The hyperparameter table is in Table 1. We ran all experiments on RTX 2080 ti GPUs. We use PyTorch 1.11 for all models, training and evaluation. We use AdamW optimizer for training. For curriculum design, we follow the graph causal relationship to reveal variables. For example, no future variables will be revealed until all of its precedents are revealed during training. For tree structures, we use BFS to determine the curriculum. We do not use any learning decay techniques as we find learned transformers perform better without it. For each training step, we generate sampled $N_{\text{train}} + 1$ examples randomly from 1 of our 50k candidate graphs to, to ensure models do not see repetitive data during training. We log training and test loss every 50 steps, and save the checkpoint with lowest training loss. For data generation, we use the Python package `pomegranate` for both constructing networks and sampling.

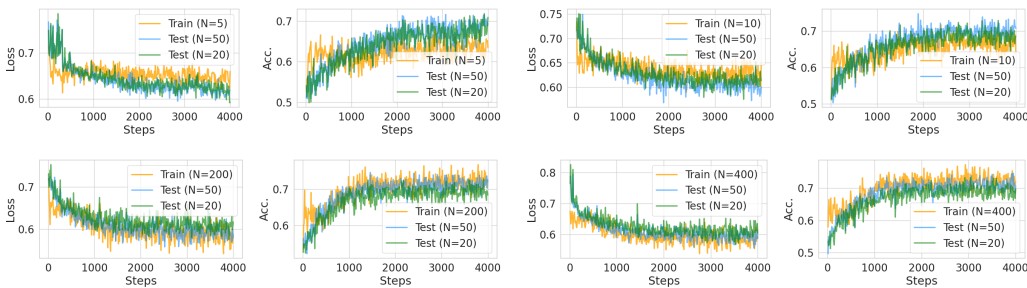

Figure 12: **Generalization Analysis: Convergence result on tree. Top:** $N_{\text{train}} \in \{5, 10\}$**, Bottom:** $N_{\text{train}} \in \{200, 400\}$ Similar to our results on graph, transformers trained on large $N_{\text{train}}$ generalize better than trained on smaller $N_{\text{train}}$. The gap between training and testset gets larger close to the end of training.

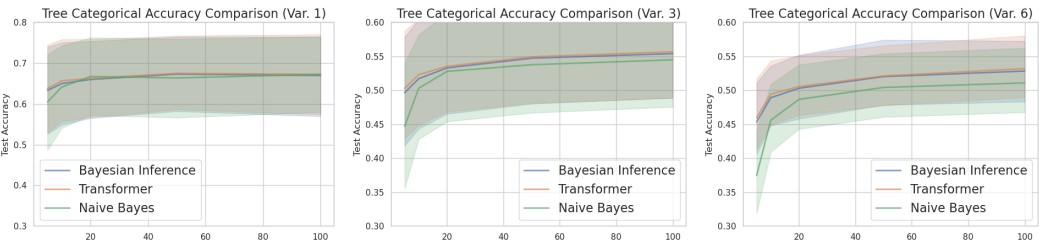

Figure 13: **Accuracy Comparison for the Tree Network with Categorical distribution.** In the figures, we are able to observe the test accuracy follows the same pattern comparing to the ones with binary distribution (Figure 2). The result shows that transformers are capable of learning the network structure and perform Bayesian inference.

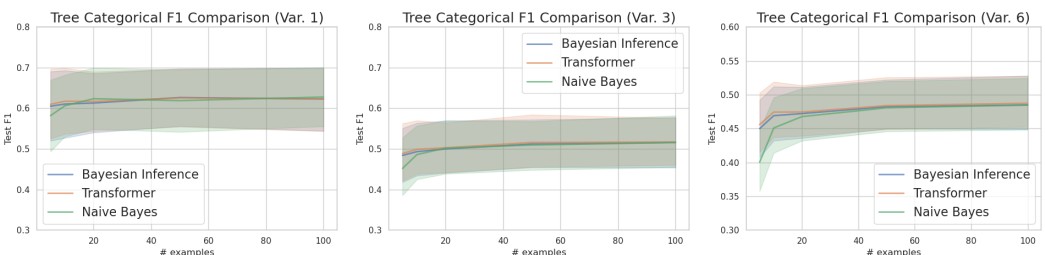

Figure 14: **F1 Score Comparison for the Tree Network with Categorical distribution.** Since we are handling the multi-class prediction, we also report the F1 score for all the baselines. Similar to what we observe in the accuracy result, we are also able to observe the test F1 follows the same pattern comparing to the ones with binary distribution (Figure 2). The result again confirms that transformers are capable of learning the network structure and perform Bayesian inference.

## C.3 BASELINES

Here we explain the baselines used in our experiments. We use an example for predicting a $M$-variable chain to explain the two baselines. Note that the two baselines are not capable of handling unseen features or labels. Such a case will lead directly to assigning probability $0$ to all categories.

**Naive Bayes.** For the naive Bayes baseline, we assume the graph structure is unknown. For instance, assuming $N$ in-context examples: $[\mathbf{x}_i]_{i=1}^N$, where $\mathbf{x}_i = (\mathbf{x}_{1i}, \ldots, \mathbf{x}_{Mi})$. To predict the $m$-th variable of the test token $\mathbf{x}_{mq}$, we consider the following conditional probability.

$$\mathbb{P}\left[\mathbf{x}_{mq} = 0\right] := \mathbb{P}\left[\mathbf{x}_{mi} = 0 \mid \mathbf{x}_{\mu i} = \mathbf{x}_{\mu q} \text{ for all } \mu \in [m-1], i \in [N]\right]. \tag{C.1}$$

**Bayesian Inference.** For Bayesian inference, we assume the graph structure is known. Therefore, following the above example, to estimate the $m$-th variable of a chain, we only consider the marginal probability conditioned on the observation of $(m-1)$-th variable (parent).

$$\mathbb{P}\left[\mathbf{x}_{mq} = 0\right] := \mathbb{P}\left[\mathbf{x}_{mi} = 0 \mid \mathbf{x}_{(m-1)i} = \mathbf{x}_{(m-1)q} \text{ for all } i \in [N]\right]. \tag{C.2}$$

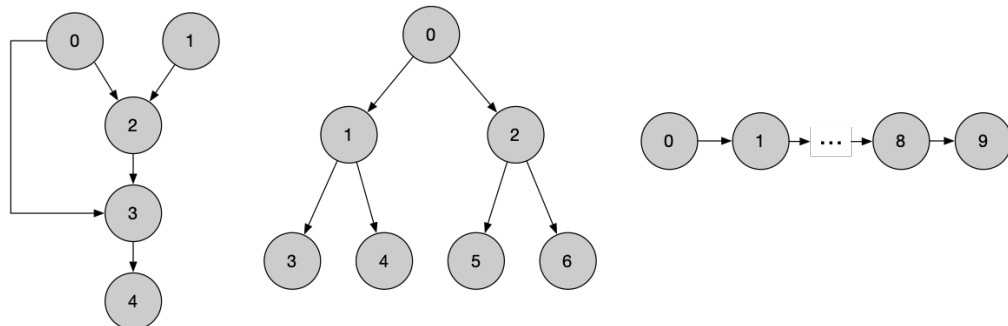

Figure 15: **Illustrations of graph structures in the experiments.** Left to right: general graph, tree and chain. The curriculum follows the number order of variables. Note that for general graph, variable 2, 3 both have 2 parents. However, for variable 2, the modeling process is identical for naive Bayes and Bayesian inference. For variable 3, modeling it is different for naive Bayes and Bayesian inference.

Table 1: Hyperparameters.

| parameter | Chain | Tree | General |
|---|---|---|---|
| optimizer | AdamW | AdamW | AdamW |
| steps | 10k | 3k | 2k |
| learning rate | 1e-4 | 5e-4 | 5e-4 |
| weight decay | 1e-2 | 5e-2 | 5e-2 |
| batch size | 64 | 64 | 64 |
| number of layers | 6 | 6 | 6 |
| loss function | Cross Entropy | Cross Entropy | Cross Entropy |
| hidden dimension | 256 | 256 | 256 |
| number of heads | 8 | 8 | 8 |
| number of examples (Train) $N$ | 100 | 100 | 100 |

