# OpenReview forum: "Transformers Learn Bayesian Networks Autoregressively In-Context"
_ICLR.cc/2025/Conference — ICLR 2025 Conference Withdrawn Submission_

### Official Review · Reviewer_WKSB · 2024-11-02

**Soundness:** 1
**Presentation:** 2
**Contribution:** 2
**Rating:** 3
**Confidence:** 3

**Summary:**

The paper studies the problem of in-context learning in transformers. In particular, it focuses on whether transformers are able to learn Bayesian networks in-context. In this setting, the model, given N different realisations of a specific graph and a query sample, is tasked to predict the probability distribution associated with a missing variable (see construction in Eq. 3.2). The assumption is that, if the model is able to infer the conditional probabilities associated with the Bayesian network, it can then use them to predict the value of the missing variable. In addition, once the model has captured such conditional probabilities, it is in principle able to generate new samples from the inferred graphical model (Algorithm 1).

The authors first provide a theoretical construction for a two-layer transformer which is capable of estimating the conditional probabilities of the Bayesian network according to the context, and autoregressively generating a new sample according to the Bayesian network with estimated conditional probabilities (Theorem 4.1 and Lemma 6.1 and Lemma 6.2).

The authors also conduct an empirical analysis to show the performance of trained transformers (with up to 6 layers) on the task of in-context learning three graph structures, namely a "general graph", a "tree" and a "chain". The performance of the model is studied by varying the number of in-context examples seen at training time and evaluating the model on different number of test in-context samples. The results show some evidence that transformers are capable of learning Bayesian networks in-context.

**Strengths:**

- The paper follows a relevant and fruitful line of work studying in-context-learning (ICL) on controlled settings.

- The paper proposes the interesting benchmark of Bayesian networks to study ICL capabilities of transformers.

- The paper provides a theoretical construction of a simple 2-layer transformers capable of estimating conditional probabilities of Bayesian networks and of generating a new sample autoregressively from the inferred graphical model.

**Weaknesses:**

- I find the notation quite confusing and the way the paper is organised makes it a bit hard to follow. For example, looking at Algorithm 3.1, it seems that the input to the model is of size (2M+1)d x (N+1) where the N+1 factor takes the query into account, while it seems the Read Function takes as input a tensor of size (2M+1)d x (N). In addition, I found a bit hard to follow the description of how the training and test datasets are generated (paragraphs Datasets and Metrics in Section 5.1). Could the authors clarify these points?

- As far as I understand, the paper focuses on only three different Bayesian networks with a fixed structure (shown in Figure. 1). If my understanding is correct, I believe more varied and diverse graph structures should be considered to better support the author's thesis. Can the analysis be extended to other graphs?

- On a related note, why only binary variables are considered? It would be interesting to extend the analysis to variables taking values from a vocabulary of a certain size.

- In section 5.1 (model paragraph), the dimensions of p and p_q changes compared to Eq. 3.1 where they were defined. Could the authors please clarify?

- From the experiments in section 5.3 a one layer transformer seems to be enough. This result contrasts with the theoretical construction which in principle would require a 2-layer model. Could the authors better elaborate on this point?

- Several typos across the manuscript. See, for example, missing link in the "Curriculum Design" paragraph ("A visualization of the curriculum is in XXX")

**Questions:**

See weaknesses part.

---

> ### Author Response · Authors · 2024-12-04
>
> Thank you very much for your helpful comments. We address your questions as follows.
>
>
> > *Q1.* I find the notation quite confusing and the way the paper is organised makes it a bit hard to follow.
>
> *A1.* Thanks for pointing it out. We believe that your confusions are caused by several typos and unexplained notations. We will fix them in the revision.
>
> >*Q2.* As far as I understand, the paper focuses on only three different Bayesian networks with a fixed structure (shown in Figure. 1).
>
> *A2.* We believe this is a misunderstanding. Our theory holds for arbitrary architectures of Bayesian networks. Our experiments consider example structures illustrated in Figure. 1 as we believe they are representative graph structures. We will add real data experiments in the revision.
>
> >*Q3.* On a related note, why only binary variables are considered? It would be interesting to extend the analysis to variables taking values from a vocabulary of a certain size.
>
> *A3.* We believe this is also a misunderstanding. Our theory applies to any discrete random variables and our theorem results are established for the case where each random variable can take $d$ different values. We have also added experiments in our revised paper on multi-category variables (Figures 13 and 14).
>
> >*Q4.* In section 5.1 (model paragraph), the dimensions of p and p_q changes compared to Eq. 3.1 where they were defined. Could the authors please clarify?
>
> *A4.* Thanks for pointing it out. We will thoroughly revise the paper and ensure consistent notations.
>
> >*Q5.* From the experiments in section 5.3 a one layer transformer seems to be enough. This result contrasts with the theoretical construction which in principle would require a 2-layer model. Could the authors better elaborate on this point?
>
> *A5.* We believe there is no contradiction between our theory and experiments. Our theory shows that there exists a simple transformer model with good performance, providing practical guidance that a well-trained transformer should at least perform similarly as the one we theoretically construct. Our theory does not deny the possibilty that there exists other transformer models that can perform equally well on the same task.
>
> >*Q6.* Typos.
>
> *A6.* Thank you for pointing out the typos. We will fix them in the revision.

---

### Official Review · Reviewer_j98B · 2024-11-03

**Soundness:** 3
**Presentation:** 2
**Contribution:** 2
**Rating:** 5
**Confidence:** 4

**Summary:**

The main goal of this paper is to demonstrate how transformers can learn Bayesian networks in context. The paper provides a theoretical proof showing that transformers are capable of modeling Bayesian networks in context. Additionally, The paper provides an evidence that transformers trained on such tasks can make Bayes-optimal predictions for new samples in an autoregressive manner.

**Strengths:**

1. The paper proposes a theoretical construction that transformers are capable to capture  Bayesian networks in context.
2. The paper presents a well-defined experimental framework to explore how transformers learn Bayesian networks in context,  which could inspire further research.
3. The paper compares prediction accuracy by varying the variables and types of Bayesian networks, providing a detailed description of qualitative differences among various instances.

**Weaknesses:**

1. The paper does not provide evidence on whether the trained transformer implements the algorithm proposed in Theorem 4.1 and Section 6. Other previous works on similar topics utilizes attention pattern analysis and causal studies through ablations.
2. The paper lacks explanations of terms like “naive bayes” and “bayesian inference” and does not clarify how the accuracy of these algorithms is calculated in the accuracy plots in Figures 2, 4, and 6.
3. The paper does not address the robustness of the results under more realistic settings, such as with positional embeddings.

**Questions:**

1. Could you specify “naive bayes” and “bayesian inference” in main text?
2. Could you provide whether the trained transformers implement an algorithm proposed at Theorem 4.1 and Section 6?

---

> ### Author Response · Authors · 2024-12-04
>
> Thank you very much for your helpful comments. We give our detailed reponses to your questions below.
>
>
> >*Q1.* The paper does not provide evidence on whether the trained transformer implements the algorithm proposed in Theorem 4.1 and Section 6. Other previous works on similar topics utilizes attention pattern analysis and causal studies through ablations.
>
> *A1.* Thank you for your question and suggestion. Please note that as a study focusing on the expressive power of transformers, the goal of our paper is to demonstrate that there exists a simple transformer capable of handling our task of interest. To our knowledge, most results of this type do not necessarily ensure that the actual model obtained through training is exactly the same as the one constructed in theory.
>
> Our theory shows that there exists a simple transformer model with good performance, providing practical guidance that a well-trained transformer should at least perform similarly as the one we theoretically construct. Therefore, even if there is no exact match between the trained model and the theoretical construction, it does not diminish the practical value of our theory.
>
>
>
> >*Q2.* The paper lacks explanations of terms like “naive bayes” and “bayesian inference” and does not clarify how the accuracy of these algorithms is calculated in the accuracy plots in Figures 2, 4, and 6.
>
> *A2.* Please refer to *A4* in our response to all reviewers.
>
>
> >*Q3.* The paper does not address the robustness of the results under more realistic settings, such as with positional embeddings.
>
> *A3.* Please note that, as a paper focusing on the expressive power of transformers, extensions of our result to settings with practical positional embeddings is trivial – we can easily slightly modify a bias term to subtract the unused positional embeddings. Positional embeddings do not significantly affect the experiment results either.

---

### Official Review · Reviewer_6Gv6 · 2024-11-03

**Soundness:** 1
**Presentation:** 2
**Contribution:** 1
**Rating:** 1
**Confidence:** 5

**Summary:**

This paper considers the ability of Transformers to estimate in-context the conditional probabilities of categorical variables. Theoretically, the paper seeks to prove that for any joint distribution over categorical variables and an ordering over them, there exists a two-layer Transformer that can represent the true conditional probabilities of each variable given those that are earlier in the ordering. Empirically, the paper considers experiments on synthetic data where Transformers are trained on samples from different Bayesian networks that all come from some family of graphs. The paper compares the probabilities estimated in-context to the ground truth as well as those estimated via naive Bayes and Bayesian inference, finding trends that suggest that Transformers have the capacity to estimate conditional probabilities in-context.

**Strengths:**

This paper investigates whether Transformers are capable of estimating multivariate discrete distributions in-context. In and of itself, this research question has not been studied yet, to the best of my knowledge.

**Weaknesses:**

+ A key flaw with this paper is a misrepresentation of the problem: this is not a paper about learning Bayesian networks in-context; rather, it is a paper about whether Transformers can estimate conditional probabilities of discrete variables in-context. In lines 106-107, where the problem is introduced, note that the Bayesian network that is specified is not the true Bayesian network (BN) that defines the joint distribution of the variables: it is simply a factorization of the distribution via chain rule given a particular variable ordering. This factorization is generic and valid for any distribution. By contrast, the __true__ BN that underlies a distribution can entail far more conditional independences than is given by the chain rule. Even if this paper was about learning BNs, BNs are anyways not identified by observed data: it is known theory that multiple BNs entail the exact same set of conditional independences.

+ Following up on the previous point, the paper can be interpreted as asking: can Transformers estimate joint distributions of discrete variables in-context? The technical result is in service of showing that true conditional probabilities can be captured by the hypothesis class of two-layer Transformers. But, the significance of this finding lacks context: what is the broader implication if Transformers can estimate multivariate discrete distributions in-context? What questions will this help us answer in the broader context of machine learning? The authors need to properly contextualize the questions and finding in their paper.

+ The technical setup lacks clarity about details that are essential to a paper about Transformers and in-context learning: what is the precise objective by which the Transformer is trained? Is it a causal decoding Transformer trained to minimize the negative log likelihood of the next categorical variable given the previous ones in a particular sample? Details about how the Transformer is trained are completely missing. Further, for completeness, the authors should also properly define every piece of notation like $0_{dm}$ and $\mathbf{e}_{N+1}$ -- I imagine these define a matrix of 0s and the $N+1$-th standard basis vector, respectively? But readers shouldn't have to interpret key pieces of notation.

+ There are technical details that do not appear to be correct. For example, in Eqn. 3.2 that defines the input matrix $\mathbf{X}$ to the Transformer, the dimensions do not make sense: each $\mathbf{x}_{ij}$ entry is a $d$-dimensional one-hot encoding, as stated in in line 117, but the vector $p$ is $(M+1)d$-dimensional according to Eqn. 3.1. Thus, the last row of the input $\mathbf{X}$ seems to have more columns than the rows above. Another example is in line 190: to specify the output of the model, the authors indicate $\mathbb{R}^d$ and define operations that would produce a $d$-dimensional real-valued vector, but for categorical variables, we need to output vectors in the $d$-dimensional simplex. The composition of the $\mathrm{Read}(\cdot)$ and $\mathrm{Linear}(\cdot)$ functions would not produce vectors that are probabilities that sum to 1, as needed for evaluating the log likelihood or for sampling discrete variables.

+ The empirical studies also lack clarity about key details. For example, in lines 261 and 262, the phrase "the probability distribution of those graphs ..." is not parseable. What is this referring to? Second, the methods that are compared with a Transformer -- naive Bayes and Bayesian inference -- are significantly lacking in clarity. How is naive Bayes being applied to the density estimation problem considered in this paper? Bayesian inference is not a model, it is a method, so what is the underlying model on which Bayesian inference is applied and what is the posterior being inferred? These details are not clear from the paper and limit the ability of a reader to make sense of the empirical findings.

**Questions:**

+ Can you please clarify the problem formulation in this paper? I don't think it's accurate to say that this paper is about Bayesian network learning. However, I'd like the authors to reflect on this aspect, and clarify this point.

+ Can you elaborate significantly on how the Transformer is trained, including details about: is it a causal decoding Transformer? Is it trained to minimize the negative log-likelihood of the next variable given previous ones? Include all details that can help a reader clearly understand the training objective.

+ Can you shed light on the dimensions of the input $X$ and in particular, clarify the apparent mismatch in dimensions of the last row against the previous rows?

+ Can you clarify how the output of the final linear layer is transformed to produce a proper vector of probabilities for a categorical distribution?

+ Can you clarify the missing details about the empirical studies that I noted in the "weaknesses" section?

---

> ### Author Response · Authors · 2024-12-04
>
> Thank you very much for your detailed comments and constructive suggestions. We address yoru questions as follows.
>
> >*Q1.* A key flaw with this paper is a misrepresentation of the problem: this is not a paper about learning Bayesian networks in-context; rather, it is a paper about whether Transformers can estimate conditional probabilities of discrete variables in-context.
>
> *A1.* Please refer to *A2* in our response to all reviewers. We will clarify in our revision that our goal is to show that there exists a simple transformer model that can (i) estimate the conditional probabilities of the Bayesian network according to the context, and (ii) autoregressively generate a new sample according to the Bayesian network with estimated conditional probabilities. We have made clarifications in the abstract, and we will consider changing the title of the paper to further avoid confusion.
>
>
> >*Q2.* Following up on the previous point, the paper can be interpreted as asking: can Transformers estimate joint distributions of discrete variables in-context? The technical result is in service of showing that true conditional probabilities can be captured by the hypothesis class of two-layer Transformers. But, the significance of this finding lacks context: what is the broader implication if Transformers can estimate multivariate discrete distributions in-context? What questions will this help us answer in the broader context of machine learning? The authors need to properly contextualize the questions and finding in their paper.
>
> *A2.* Thank you very much for your suggestions. We do agree that our paper needs a better explanation of the background and we will focus on it in our revision.
>
> >*Q3.* The technical setup lacks clarity about details that are essential to a paper about Transformers and in-context learning: what is the precise objective by which the Transformer is trained? Is it a causal decoding Transformer trained to minimize the negative log likelihood of the next categorical variable given the previous ones in a particular sample?
>
> *A3.* Please refer to *A3* in our response to all reviewers.
>
> >*Q4.* There are technical details that do not appear to be correct. For example, in Eqn. 3.2 …
>
> *A4.* Thanks for pointing it out. We believe your confusions are caused by several typos and unexplained notations. We will fix them in the revision.
>
> >*Q5* The empirical studies also lack clarity about key details. For example, in lines 261 and 262,...
>
> *A5.* We will thoroughly revise the paper to improve the presentation. Regarding 'naive Bayes' and 'Bayesian inference', please refer to *A4* in our response to all reviewers.

---

### Official Review · Reviewer_WQ4Q · 2024-11-04

**Soundness:** 3
**Presentation:** 3
**Contribution:** 3
**Rating:** 5
**Confidence:** 3

**Summary:**

This paper theoretically constructs a simple transformer model that learns to sample from a Bayesian Network (BN) from in-context samples. A BN is an ordered set of variables that has a causal ordering among their variables and satisfy various conditional probabilities. The paper shows that for a BN of bounded maximum indegree, a simple 2-layer transformer can approximate the conditional probabilities and generate new samples. The proof is simple and basically constructs positional embeddings that mimic the parent structure of the BN and then applies MLE. Experiments are conducted to validate the theory on simulated BNs and also probe the number of layers needed. The target audience are people interested in theoretical machine learning.

**Strengths:**

- There has been a lot of growing interest in theoretically studying whether transformers can learn certain structures [1, 2, 3]. The problem this work studies, whether transformers learn bayesian networks, is very interesting and relevant to the ICLR community.

- The general problem of learning a BN is very tricky (even with transformers) and the work simplifies it nicely using a curriculum approach so only a few variables are introduced at each stage. However, while the idea is novel, this does limit the usefulness of this algorithm (see weaknesses below).

#### References:

- [1] Transformers Learn Shortcuts to Automata
- [2] Do LLMs dream of elephants (when told not to)? Latent concept association and associative memory in transformers.
- [3] (Un)interpretability of Transformers: a case study with bounded Dyck grammars

**Weaknesses:**

- While the result is nice to have, it's unclear whether how the main theorem of this work compares to results from existing works on universal approximation capabilities of transformers.

- Moreover, it's also unclear whether gradient descent or standard approximation methods used to learn such models will extract some sort of similar structure. The authors state this in their conclusion, however this is a relevant weakness of this work and limits its utility.

- The curriculum setup sounds interesting, however it seems to require apriori knowledge of the causal order and this may not be available in practice.

- While experiments on simulated data validate the theory, it would also be nice to have some validation on real-life data (even on few variables).

**Questions:**

Some questions were raised above.

- In the definition of a BN, the causal order seems to respect the index order. Does the main theorem hold when the ordering is not known, i.e. the variables are permuted uniformly in the samples?

#### Typos:

- L152: will-known -> well-known
- L198: paramter -> parameter
- L294: missing citation for visualization
- L667: nad -> and

---

> ### Author Response · Authors · 2024-12-04
>
> Thank you very much for your helpful comments. We address your questions as follows.
>
> > *Q1.* While the result is nice to have, it's unclear how the main theorem of this work compares to results from existing works on universal approximation capabilities of transformers.
>
> *A1.* Thanks for your question. Please note that our paper considers an in-context learning task which is rather complicated. You are correct that existing results on universal approximation capabilities of transformers cannot imply any concrete results in our setting, and this is the strength of our results.
>
>
> > *Q2.* Moreover, it's also unclear whether gradient descent or standard approximation methods used to learn such models will extract some sort of similar structure. The authors state this in their conclusion, however this is a relevant weakness of this work and limits its utility.
>
> *A2.* While our theory does not demonstrate whether standard training algorithms can indeed give a transformer model that can accomplish our desired tasks, we demonstrate this through experiments. We also propose a particular training method based on curriculum, which improves the utility of our results.
>
>
> > *Q3.* The curriculum setup sounds interesting, however it seems to require a priori knowledge of the causal order and this may not be available in practice. In the definition of a BN, the causal order seems to respect the index order. Does the main theorem hold when the ordering is not known, i.e. the variables are permuted uniformly in the samples?
>
>
> *A3.* Please refer to *A1.* in our response to all reviewers. We believe such a setting is natural given that we are considering an autoregressive task. The order we consider does not have to be the ‘causal’ order. It can be any order of variables (and there always exist a Baysian network following this order that can describe the joint distribution of the random variables).
>
>
> > *Q4.* While experiments on simulated data validate the theory, it would also be nice to have some validation on real-life data (even on few variables).
>
> *A4.* Thank you for your suggestion. We will add real data experiments in our revised paper.
>
> > *Q5.* Typos.
>
> *A5.* Thanks for pointing out the typos. We will fix them in the revision.

---

### Author Response · Authors · 2024-12-04
**Response to All Reviewers**

Dear Reviewers,

Thank you very much for your constructive and helpful comments. We realize that many of your concerns are caused by unclear terminologies and insufficient background explanations, and that our work can benefit from a thorough revision. Therefore, we decide to withdraw our submission. We will carefully revise the paper based on your valuable feedback before submitting it to future venues. We would like to respond to your major concerns as follows.

> *Q1.* Our setting requires a priori knowledge of the causal order of variables.

*A1.* Our paper aims to demonstrate that transformers can *autoregressively* generate new samples according to an estimated Bayesian network. Having a pre-determined order of the variables is natural and is consistent with the practice, since we consider autoregressive generation.

Please also note that without a pre-determined order of variables, Bayesian networks are not identifiable because multiple Bayesian networks can equivalently describe the same joint distribution of variables. Specifically, it can be shown that for any given order of variables, there exists a Bayesian network that describes the same joint distribution, satisfying that each variable can only be a descendant of the preceding variables according to that order.

> *Q2.* This is not a paper about learning Bayesian networks in-context; rather, it is a paper about whether Transformers can estimate conditional probabilities of discrete variables in-context.

*A2.* Thanks for pointing this out. We will clarify in our revision that our goal is to show that there exists a simple transformer model that can (i) estimate the conditional probabilities of the Bayesian network according to the context, and (ii) autoregressively generate a new sample according to the Bayesian network with estimated conditional probabilities. We have made clarifications in the abstract, and we will consider changing the title of the paper to further avoid confusion.

Please note that similar settings of estimating the conditional probabilities of the Bayesian network according to the context is standard and has been considered in recent work [1].

> *Q3.* What is the precise objective by which the Transformer is trained?

*A3.* Our theory focuses on studying the expressive power of transformers. The nature of this kind of research is that it does not necessarily focus on any particular training objective. Instead, we just aim to show that there exists such a transformer model that can accomplish the desired task. Similar studies of the expressive power of transformers are also considered in previous works such as [2].

In our experiments, we train the transformers to minimize the cross-entropy loss of predicting the masked-out variables in queries. We will add clarifications to this together with a more detailed explanation of the curriculum.


> *Q4.* Explanations of the 'naive Bayes' and 'Bayesian inference' methods we used for performance comparison with transformers in our experiments.

*A4.* We have realized that these are confusing terminologies, and we will replace them with clearer names in the revision.

Suppose that there are $M$ variables $X_1,\ldots,X_M$, and that we have $N$ independent groups of observations $(X\_{11},\ldots,X\_{M1}),\ldots,(X\_{1N},\ldots,X\_{MN})$. Further suppose that we have query observations $X\_{1q},\ldots,X\_{(m_0-1)q}$, and our goal is to estimate the conditional probabilities of the form:

$P( X\_{m_0} = j | X\_{1} = X\_{1q},\ldots, X\_{m_0-1} = X\_{(m_0-1)q} )\quad\quad (Eq1) $.

Denote by $\mathcal{P}(m_0) $ the parent set of the $m_0$-th variable according to the Bayesian network. In our current manuscript, we mean by 'naive Bayes' the method that estimates the conditional probability above in (Eq1) as

$ \frac{ | \{ i\in [N]: X\_{m_0i} = j, \text{ and } X\_{mi} = X\_{mq} \text{ for all } m = 1,\ldots,m_0-1 \} | }{ | \{ i\in [N]: X\_{mi} = X\_{mq} \text{ for all } m = 1,\ldots,m_0-1 \} | }$.

We mean by ‘Bayesian inference' the method that estimates the conditional probability above in (Eq1) as

$ \frac{ | \{ i\in [N]: X\_{m_0i} = j, \text{ and } X\_{mi} = X\_{mq} \text{ for all } m\in \mathcal{P}(m_0) \} | }{ | \{ i\in [N]: X\_{mi} = X\_{mq} \text{ for all } m\in \mathcal{P}(m_0) \} | } $.

Clearly, 'Bayesian inference' can utilize more observations to calculate frequencies and is therefore more efficient. According to our theory, the performance of transformers should be comparable to 'Bayesian inference' and better than 'naive Bayes', particularly when the ground truth Bayesian network is complex.

We hope that our response above can address most of your concerns. Thank you again for your valuable feedback.

Best regards,

Authors

[1] Eshaan Nichani, et al. "How Transformers Learn Causal Structure with Gradient Descent." ICML 2024.

[2] Yu Bai, et al. "Transformers as statisticians: Provable in-context learning with in-context algorithm selection." NeurIPS 2023.

---

### Note · Authors · 2024-12-04

**Comment:**

We thank the ACs for handling our paper and thank all the reviewers for the constructive and helpful comments. We realize that many of the reviewers' concerns are caused by unclear terminologies and insufficient background explanations, and that our work can benefit from a thorough revision. Therefore, we decide to withdraw our submission. We will carefully revise the paper based on the valuable feedback we receive, and submit the work to a future venue.

**Withdrawal Confirmation:**

I have read and agree with the venue's withdrawal policy on behalf of myself and my co-authors.